# QINCO2: Vector Compression and Search with Improved Implicit Neural Codebooks

**Théophane Vallaeys, Matthew Muckley, Jakob Verbeek, Matthijs Douze,**
FAIR, Meta
{`webalorn,mmuckey,jjverbeek,matthijs`}@meta.com

## Abstract

Vector quantization is a fundamental technique for compression and large-scale nearest neighbor search. For high-accuracy operating points, multi-codebook quantization associates data vectors with one element from each of multiple codebooks. An example is residual quantization (RQ), which iteratively quantizes the residual error of previous steps. Dependencies between the different parts of the code are, however, ignored in RQ, which leads to suboptimal rate-distortion performance. QINCo recently addressed this inefficiency by using a neural network to determine the quantization codebook in RQ based on the vector reconstruction from previous steps. In this paper we introduce QINCo2 which extends and improves QINCo with (i) improved vector encoding using codeword pre-selection and beam-search, (ii) a fast approximate decoder leveraging codeword pairs to establish accurate short-lists for search, and (iii) an optimized training procedure and network architecture. We conduct experiments on four datasets to evaluate QINCo2 for vector compression and billion-scale nearest neighbor search. We obtain outstanding results in both settings, improving the state-of-the-art reconstruction MSE by 34% for 16-byte vector compression on BigANN, and search accuracy by 24% with 8-byte encodings on Deep1M.

## 1 Introduction

Vector quantization is a fundamental technique, with widespread use cases from exploratory data analysis and visualization, to self-supervised learning (Caron et al., 2018), image compression (Esser et al., 2021; Careil et al., 2024) and large-scale nearest neighbor search (Jégou et al., 2010). To learn a vector quantizer from data, the k-means algorithm (Bishop, 2006; MacQueen, 1967) is probably the most ubiquitous. It associates each data vector with the nearest element in a learned codebook of centroids in the sense of mean squared error (MSE).

In data compression the goal is to optimize the rate-distortion tradeoff (Shannon, 1948). With k-means the quantization error can be reduced by using a larger number of centroids $K$, which naturally increases the bitrate of $\lceil \log_2 K \rceil$ to encode the index of the centroid assigned to a particular data vector. In practice, as the number of parameters and computational cost of the k-means grows linearly in the number of centroids, k-means is hard to scale beyond, say, 1M centroids. To enable lower distortion operating points, *multi-codebook quantization* (MCQ) methods associate each data vector with one element across each of $M > 1$ codebooks. Examples include Product Quantization (PQ), which slices data vectors in several pieces and applies k-means to each of them, and residual quantization (RQ), which iteratively applies k-means to the residual of previous quantization steps.

A fundamental limitation of both PQ and RQ is that the $M$ codebooks are pre-determined and independent of the vector to be quantized. This is suboptimal, as in general there are dependencies between different parts of the code, and one part of a PQ or RQ code carries information about other parts of the code. Such dependencies can be modeled using an entropy model, as *e.g.* explored by El-Nouby et al. (2023) for image compression, but this only reduces the bitrate and does not directly address the inherent inefficiency of the quantizer. Huijben et al. (2024) introduced QINCo, a neural variant of RQ, that reduces this redundancy where each codebook is computed as a function of earlier selected codes. QINCo is initialized from RQ codebooks and learns a neural network to update the codewords for the current step as a function of the vector reconstruction from previous steps. This leads to large improvements in MSE reconstruction and nearest neighbor search accuracy.

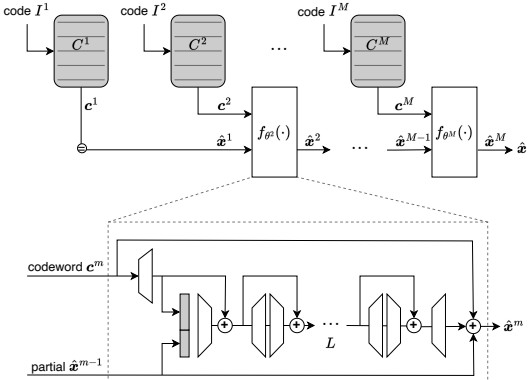

Figure 1: **Overview of the QINCO2 architecture, as used during decoding.** Top: The codes $(i^1, \ldots, i^M)$ are mapped to codewords $(c^1, \ldots, c^M)$ using look-up tables $C^1, \ldots, C^M$. These codes are sequentially combined using a parameterized function $f_{\theta^m}$. Bottom: The function $f_{\theta^m}$ applied on a codeword $c^m$ and the partial reconstruction $\hat{x}^{m-1}$ from the previous step, using a sequence of residual MLPs.

In this paper we introduce QINCO2 which extends and improves upon QINCO in several ways. (i) We improve the vector encoding using codeword pre-selection and beam-search: pre-selection reduces the number of function evaluations to update RQ centroids which makes beam search more affordable, significantly reducing the quantization error. (ii) We introduce a fast, look-up based, approximate decoder for the QINCO2 codes to establish shortlists for large-scale search: the decoder works akin to RQ but uses a pair of QINCO2 codes in each step to account for their dependencies. (iii) We optimize the training procedure and network architecture: this includes a configurable internal feature dimension of the network and an additional residual connection, improved initialization and larger batch size. See Figure 1 for an illustration of the QINCO2 architecture.

We conduct extensive experiments on four different common vector compression datasets: Deep1B, BigANN, FB-ssnpp, and Contriever. We evaluate QINCO2 for vector compression performance in terms of reconstruction MSE and nearest neighbor accuracy on 1M sized databases. In addition, we evaluate in experiments in a billion-scale vector search setting, in terms of the search-speed *vs* accuracy trade-off. We obtain outstanding results with QINCO2 in both settings. For example, compared to the recent state-of-the-art QINCO approach, we reduce reconstruction MSE by 34% from $0.32 \times 10^{-4}$ to $0.18 \times 10^{-4}$ for 16-byte compression of vectors in the BigANN dataset, and improved nearest neighbor accuracy by 24% from 36.3% to 45.1% for 8-byte compression of vectors in the Deep1M dataset. Code is available at: https://github.com/facebookresearch/Qinco

## 2 BACKGROUND AND RELATED WORK

**Multi-codebook quantization.** This family of methods builds on multiple k-means sub-quantizers, using different codebooks. With $M$ sub-quantizers, the total code size becomes $M\lceil \log_2(K) \rceil$ bits. Product Quantization (PQ) (Jégou et al., 2010) splits the vectors into $M$ sub-vectors that are quantized separately. As the sub-codes are estimated independently, finding the optimal encoding is trivial. In Residual Quantization (RQ) (Liu et al., 2015) k-means quantizers are applied sequentially such that each quantizer encodes the residual left from the previous step, and decoding sums the selected $M$ codewords. RQ encoding is greedy and therefore potentially suboptimal. It can, however, be improved using beam search to explore several encodings in parallel. In Additive Quantization (AQ) (Babenko & Lempitsky, 2014) the $M$ sub-codes are estimated simultaneously for encoding, while using the same sum-of-codewords decoding as RQ. LSQ (Martinez et al., 2018) is a state-of-the-art AQ variant that relies on annealed optimization, where the encoding accuracy depends on the number of optimization iterations. RVPQ (Niu et al., 2023) combines PQ with RQ by slicing the vectors in a number of components, and applying RQ with a beam search on each of them.

**Neural vector quantization.** Vector quantization has been used in neural networks for different purposes. For generative modeling, VQ-VAE (Oord et al., 2017; Razavi et al., 2019; Esser et al., 2021) uses single-step vector quantization in the latent space of a variational autoencoder (Kingma & Welling, 2014), and generates samples from an autoregressive sequence model fitted to the quantization indices. RQ-VAE (Lee et al., 2022) relies instead on residual quantization to better approximate the latents, while RAQ-VAE (Seo & Kang, 2024) enables rate-adaptive quantization within VQ-VAE with a sequence-to-sequence model. In a similar spirit, El-Nouby et al. (2023) uses a transformer-based entropy model to reduce the bitrate for image compression. Careil et al. (2024) uses a quantized encoder, and couple it with a diffusion-based decoder for high-realism image com-

pression. Closer to our work, UNQ (Morozov & Babenko, 2019) considers vector compression and jointly trains an encoder-decoder model with multiple codebooks in the latent space, using a straight-through Gumbel-Softmax estimator for differentiation. DeepQ (Zhu et al., 2023) similarly uses an encoder that maps the input vector to a set of independently sampled indices, but relies on a REINFORCE gradient estimator, simply summing the selected codewords rather than using a decoder network. In our work, we do not use an encoder-decoder pair to map the data to a latent space for quantization. Instead, we directly quantize the data in the data space, thereby avoiding any data loss incurred by the encoder-decoder, and using a residual neural quantizer to encode the data. QINCo (Huijben et al., 2024) is an improved residual quantizer that uses neural networks to adjust the quantization codebooks for each data vector, based on the reconstruction obtained in previous steps. In our work, we build upon QINCo and improve its performance by optimizing the network architecture and training procedure, introduce fast codeword selection in encoding to make beam search more affordable.

**Large-scale nearest neighbor search.** Vector quantization is a fundamental component of approximate search methods. Rather than using exhaustive search, a promising subset of the database is determined using a coarse quantization. An inverted file index (IVF) is used to store which part of the data is present in each cluster (Jégou et al., 2010), and efficient algorithms such as HNSW (Malkov & Yashunin, 2018) determine which clusters should be accessed given a query. Finally, the query is matched with the (quantized) data in the identified clusters. As decoding with QINCo is expensive, Huijben et al. (2024) proposed an additional step based on an efficient approximate additive decoder, to reduce the amount of data that needs to be passed through the neural decoder. We improve upon this approach by introducing an additive decoder that leverages the dependencies of *pairs of codewords* to boost the shortlist accuracy, allowing to reduce their size and improve efficiency. Amara et al. (2022) considered multi-layer neural network decoders to improve the accuracy of additive linear decoders, but keep the codes themselves fixed. Our pairwise decoder similarly provides an alternative decoder for given codes, but in our case we seek a less accurate but faster decoder.

## 3 IMPLICIT NEURAL CODEBOOKS

### 3.1 NOTATION AND BACKGROUND

**Multi-codebook quantization.** We start by introducing notations describing the general framework of multi-codebook quantization. We denote the vectors that we aim to quantize as $\boldsymbol{x} \in \mathbb{R}^d$, which follow an unknown distribution, accessed only via data sampled from it. The quantization process is characterized by a set of $M$ codebooks $C^1, \ldots, C^M$ with $K$ elements each and parameterized by $\theta$, where $C^m = \{\boldsymbol{c}_1^m, \ldots, \boldsymbol{c}_K^m\}$, and by a decoding function $F(\boldsymbol{c}^1, \ldots, \boldsymbol{c}^M)$. Its error is measured by a loss function, for which we consider the $\ell_2$ MSE loss defined by $\mathcal{L}(\boldsymbol{x}, \boldsymbol{q}) = \|\boldsymbol{x} - \boldsymbol{q}\|_2^2$. We define a general quantization procedure as

$$\mathcal{Q} : \boldsymbol{x} \mapsto \underset{\boldsymbol{c}^1 \in C^1, \ldots, \boldsymbol{c}^M \in C^M}{\arg\min} \mathcal{L}(\boldsymbol{x}, F(\boldsymbol{c}^1, \ldots, \boldsymbol{c}^M)). \quad (1)$$

During training, the objective is to find parameters $\theta$ minimizing the expected reconstruction loss: $\mathbb{E}_{\boldsymbol{x}} [\mathcal{L}(\boldsymbol{x}, \mathcal{Q}(\boldsymbol{x})]$. The quantized vectors can then be stored with indices using $M \lceil \log_2 K \rceil$ bits.

Residual Quantization (RQ) (Chen et al., 2010) defines $F_\oplus(\boldsymbol{c}^1, \ldots, \boldsymbol{c}^M) = \sum_{m=1}^M \boldsymbol{c}^m$, and couples this decoder with an approximate but efficient sequential encoding procedure $\mathcal{Q}_{\text{RQ}}$ where:

$$\hat{\boldsymbol{x}}^0 = \boldsymbol{0}, \qquad \hat{\boldsymbol{x}}^m = F_\oplus(\boldsymbol{c}^1, \ldots, \boldsymbol{c}^m), \quad (2)$$

$$\boldsymbol{c}^m = \underset{\boldsymbol{c} \in C^m}{\arg\min} \mathcal{L}(\boldsymbol{x}, F_\oplus(\boldsymbol{c}^1, \ldots, \boldsymbol{c}^{m-1}, \boldsymbol{c})), \quad (3)$$

in which $\hat{\boldsymbol{x}}^m$ is the *partial reconstruction* until stage $m$, and $\boldsymbol{r}^m = \boldsymbol{x} - \hat{\boldsymbol{x}}^{m-1}$ is the *residual* at step $m$. The codeword $\boldsymbol{c}^m$ is selected to best approximate the residual as $\boldsymbol{c}^m = \arg\min_{\boldsymbol{c} \in C^m} \mathcal{L}(\boldsymbol{r}^m, \boldsymbol{c})$.

**Quantization with implicit neural codebooks.** QINCo (Huijben et al., 2024) builds on RQ by redefining the decoding function $F$ without changing the quantization process $\mathcal{Q}_{\text{RQ}}$. The key insight is that in RQ the distribution of the residuals $\boldsymbol{r}^m$ depends on $\boldsymbol{c}^1, \ldots, \boldsymbol{c}^{m-1}$, yet all residuals are quantized using the same codebook $C^m$. In theory, one could improve this by using a codebook hierarchy $C^m(\boldsymbol{c}^1, \ldots, \boldsymbol{c}^{m-1})$ that depend on previously selected codewords, but this leads to an

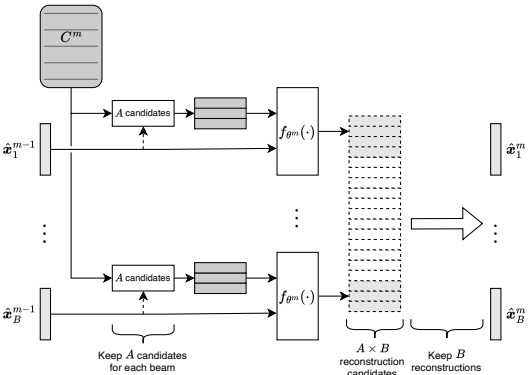

Figure 2: **Overview of the encoding process of QINCo2.** At a given step $m$, the $B$ beam search hypotheses are each combined with $A$ pre-selected candidates, for which codebook elements are computed with $f_{\theta^m}$, and the best $B$ hypotheses are retained for the next encoding step.

exponential number codebooks, which is unfeasible in practice. Instead, QINCo parameterizes $F$ using a neural network as

$$F_{\text{QI}}(\boldsymbol{c}^1, \ldots, \boldsymbol{c}^M) = \sum_m f_{\theta^m}(\boldsymbol{c}^m | \boldsymbol{x}^{m-1}), \tag{4}$$

where the $f_{\theta^m}$ are modeled by the same neural network, but with separate weights $\theta^m$ for each step. By choosing an appropriate residual architecture for the network, and initializing from RQ codebooks, the reconstruction error can be guaranteed to be no worse than that of the RQ initialization.

### 3.2 IMPROVED IMPLICIT NEURAL CODEBOOKS: ENCODING

We identify the encoding efficiency as the major bottleneck to deploy QINCo. We propose (i) a mechanism to speed it up, and (ii) the use of beam search to make it more accurate. We then (iii) adjust the architecture and training procedure to further improve quantization accuracy and training speed. We use QINCo2 to refer to models benefiting from all these improvements.

**Pre-selection.** The RQ quantization process $\mathcal{Q}_{\text{RQ}}$ applied to QINCo can be expressed as a sequence of $M$ steps, where each one consists in finding

$$\boldsymbol{c}^m = \underset{\boldsymbol{c} \in C^m}{\arg\min} \, \mathcal{L}(\boldsymbol{r}^m, f_{\theta^m}(\boldsymbol{c} | \hat{\boldsymbol{x}}^{m-1})). \tag{5}$$

This process requires $K$ evaluations of the neural network $f_{\theta^m}$, the complexity of which therefore scales linearly with the codebook size $K$. To reduce the computational cost, we propose a two-step encoding. First, we select a set $\mathcal{A}_m$ of $A$ candidates using an efficient function $g_{\phi^m}$. This function relies on a distinct codebook $\tilde{C}^m = \{\tilde{\boldsymbol{c}}_1^m, \ldots, \tilde{\boldsymbol{c}}_K^m\}$. We then compute $f_{\theta^m}$ over this restricted set, yielding the following quantization procedure which replaces Eq. (5) and we refer to as $\mathcal{Q}_{\text{QI-A}}$:

$$\mathcal{A}_m = \text{Top}_A \underset{1 \leqslant k \leqslant K}{\arg\min} \, \mathcal{L}(\boldsymbol{r}^m, g_{\phi^m}(\tilde{\boldsymbol{c}}_k^m | \hat{\boldsymbol{x}}^{m-1})), \tag{6}$$

$$\boldsymbol{c}^m = \underset{k \in \mathcal{A}_m}{\arg\min} \, \mathcal{L}(\boldsymbol{r}^m, f_{\theta^m}(\boldsymbol{c}_k^m | \hat{\boldsymbol{x}}^{m-1})). \tag{7}$$

In our experiments, $g$ uses the same architecture as $f$ with a much smaller depth $L_s$ and hidden dimension $d_{\text{h}} = 128$. In particular, with $L_s = 0$ we instead define $g(\boldsymbol{c}|\boldsymbol{x}) = \boldsymbol{c}$, which yields $\mathcal{A}_m = \text{Top}_A \arg\min_{1 \leqslant k \leqslant K} \mathcal{L}(\boldsymbol{r}^m, \tilde{\boldsymbol{c}}_k^m)$. This setting is significantly more efficient than $L_s \geqslant 1$.

**Beam search.** In RQ, instead of maintaining a single partial reconstruction across encoding steps, beam search (Babenko & Lempitsky, 2014) maintains a set of $B$ partial encodings. At quantization step $m$, each partial encoding is combined with each of the $K$ codebook elements, creating $K \times B$ new partial encodings, and the best $B$ ones are selected for the next step. Thus, one step of beam search here requires $K \times B$ evaluations of $f$, increasing compute by a factor $B$ w.r.t. greedy search. Candidate pre-selection reduces this number to $A \times B$, defining encoding process $\mathcal{Q}_{\text{QI-B}}$, at the cost of adding $K \times B$ evaluations of $g$ and of the loss function $\mathcal{L}$, whereas the greedy search QINCo baseline does $K$ evaluations of $f$. Therefore, depending on the setting of $A, B$, we can benefit from the improved search accuracy of beam search at a controlled cost. See Figure 2 for an illustration.

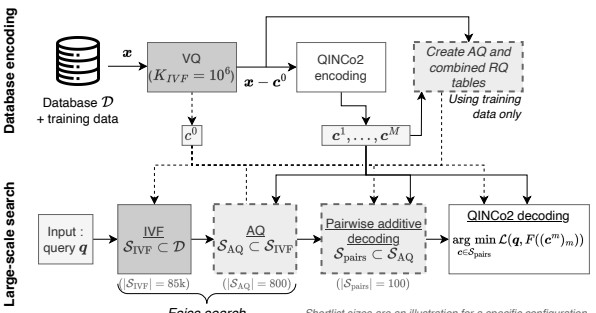

Figure 3: **Overview of our large-scale search pipeline.** We combine the compression accuracy of QINCO2 with efficient look-up methods to build and filter shortlists of candidates. Top part shows the encoding of the database and the creation of a fast searchable index. Bottom part shows the retrieval process for a given query $q$.

**Architecture.** The function $f_\theta$ in QINCO2 (see Figure 1) is similar to that of QINCO. The main differences are (i) the network backbone has dimension $d_e$, independent from vector dimension $d$ (we add linear projections between $\mathbb{R}^d$ and $\mathbb{R}^{d_e}$ at the extremities of the network); and (ii) we add a connection from the input to the output to preserve accuracy when $d_e < d$. This architecture uses $\mathcal{O}(M(Ld_e d_h + dd_e))$ FLOPs for decoding, and $\mathcal{O}(ABM(Ld_e d_h + dd_e) + BKd)$ for encoding.

**Training.** Compared to QINCO, we improve the initialization of the network and codebooks weights, the dataset normalization, the optimizer and the learning rate scheduler, and increase the batch size. We also stabilize the training by adding gradient clipping, and reduce the number of dead codewords by resetting unused ones similar to Zheng & Vedaldi (2023). Additionally, we notice that large volumes of training data are usually available for unsupervised task such as compression. Huijben et al. (2024) showed that more training data is beneficial to the accuracy of QINCO. Motivated by this observation, we train our models on the full training set of each benchmark (up to 100s of millions of vectors, see Table 1). Details of the training procedure can be found in App. A.2.

### 3.3 Large-scale nearest neighbor search

For approximate search, we build an index structure over a database of vectors $\mathcal{D} = \{x_n : n \in [\![N]\!]\}$, where $[\![N]\!] = \{1, 2, \ldots, N\}$. Given a query $q$, we search for its nearest neighbor in the $\ell_2$ sense in $\mathcal{D}$. In practice, exhaustive search in large databases is costly, and search is performed over a quantized version of the database. Overall search pipeline is shown in Figure 3.

**Creating shortlists with IVF and AQ.** Quantization reduces storage requirements, and can be leveraged to reduce the computational requirements for approximate nearest-neighbor search with QINCO2 by efficiently creating "shortlists" of the most promising database vectors. Following Huijben et al. (2024), we use IVF (Jégou et al., 2010) to group the database vectors into $K_{IVF}$ buckets $(\mathcal{U}_i)_{i \in [\![K_{IVF}]\!]}$ before quantizing the database with QINCO2, as shown in the "database encoding" (top) row of Figure 3. This is equivalent to encoding $x$ as $(I^0, I^1, \ldots, I^M)$, with the additional index $I^0 \in [\![K_{IVF}]\!]$ indexing the IVF bucket, and $I^1, \ldots, I^M \in [\![K]\!]$. The reconstruction is $\hat{x} = F(C^0(I^0), \ldots, C^M(I^M))$. For simplicity, we exclude the IVF codeword $I^0$ from the beam search.

Given a query, IVF forms $\mathcal{S}_{IVF} \subset \mathcal{D}$ from the contents of the $N_{probe} \ll K_{IVF}$ nearest buckets to the query. The size $|\mathcal{S}_{IVF}|$ of this subset ranges from a few thousands to a few millions. This is too large to rank with QINCO2 efficiently, so we re-interpret the indices $(I^1, \ldots, I^M)$ as additive quantizer codes from which approximate distances can be computed efficiently (Huijben et al., 2024). The AQ codebooks are estimated by solving a least-squares system on vectors and their corresponding codes from QINCO2 (Amara et al., 2022). This yields a second shortlist $\mathcal{S}_{AQ} \subset \mathcal{S}_{IVF}$. Only the vectors in $\mathcal{S}_{AQ}$ are then decoded using IVF-QINCO2, and ranked by their distance to the query to provide the final search result. This is represented by the "Faiss search" and "QINCO2 decoding" blocks in the bottom row of Figure 3.

**Pairwise additive decoding.** Although fast, AQ decoders trained on QINCO2 codes have a much lower search accuracy than full QINCO2 decoding. This is because the AQ decoder can only sum up independent AQ codebook entries, ignoring the dependency structure between the code elements.

We notice that AQ codebooks can be trained from any sequence of codes, including combined codes and repeated codes. We consider building a decoder based on pairs of codes. To this end we use the mapping $I^{i,j} = (I^i - 1) \times K + I^j$, with $I^i, I^j \in [\![K]\!]$ and $I^{ij} \in [\![K^2]\!]$. Combining codes in this way

makes the codebooks larger ($K^2$ instead of $K$) so they are slower to train, but also more expressive. Note that this decoder is guaranteed to be at least as good as the unitary decoder, since the codebook entries from two unitary codebooks can be combined into a pairwise decoding codebook.

The most straightforward way to take advantage of joint codebooks is to map a sequence of unitary codes $(I^1, I^2, \ldots, I^M)$ to a sequence of pairwise codes $(I^{1,2}, I^{3,4}, \ldots, I^{M-1,M})$ of length $M/2$. Associated codebooks $C'_1, \ldots, C'_{M/2}$ are creating by solving a least-squares problem using these fixed codes. However, combining the $M$ original codes into $M/2$ does not exploit all the degrees of freedom that the scheme offers. We search for a more general subset of size $M'$ of all $M(M-1)/2$ possible pairs. Inspired from RQ, given a sequence of codes $(I^1, I^2, \ldots, I^M)$ for $\boldsymbol{x} \in \mathbb{R}^d$ and a number $M'$ of target codebooks, we recursively minimize the residuals left from previous steps by searching which pairs of codes $(i, j)$ to combine and their combined codebook $C'$:

$$(i^m, j^m, C'_m) = \underset{i,j,C'}{\arg\min} \, \mathbb{E}_{\boldsymbol{x}} \left[ \mathcal{L}(\boldsymbol{r}^m, C'[I^{i,j}(\boldsymbol{x})]) \right], \tag{8}$$

$$\hat{\boldsymbol{x}}^0 = \boldsymbol{0}, \qquad \hat{\boldsymbol{x}}^m = \hat{\boldsymbol{x}}^{m-1} + C'_m[I^{i^m, j^m}(\boldsymbol{x})], \qquad \boldsymbol{r}^m = \boldsymbol{x} - \hat{\boldsymbol{x}}^{m-1} \tag{9}$$

where the expectation over $\boldsymbol{x}$ is taken w.r.t. the empirical distribution of the training data. In this approach, some input codes can be used several times, or not at all. Note that, as in RQ, the codebooks $C'_m$ are determined sequentially.

**Integration of pairwise additive decoding with IVF.** When using a pairwise additive decoder for QINCo2 without IVF, we observe that the first, and therefore most informative, code pairs are of the form $(1, i)$ or $(i, i+1)$, and then progressively include codes $(2, i)$, then $(3, i)$, *etc.* The IVF codebook of size $K_{\mathrm{IVF}}$, however, is too large to combine with other codes in our efficient pairwise decoder, since the combined codebook would be of size $K \times K_{\mathrm{IVF}} \gg K^2$. Therefore, we quantize the IVF codewords using RQ into $\tilde{M}$ new codebooks $\tilde{C}^1, \ldots, \tilde{C}^{\tilde{M}}$ each of size $K$, generating the new codes $\tilde{I}^1(\boldsymbol{x}), \ldots, \tilde{I}^{\tilde{M}}(\boldsymbol{x})$. We choose $\tilde{M}$ large enough to obtain near zero error, as the cost of increasing $\tilde{M}$ is negligible for decoding. As we only quantize the IVF codewords, we do not need to store these codes for each $\boldsymbol{x}$, but only a mapping from $I^0$ to $\tilde{I}^1, \ldots, \tilde{I}^n$, which has a size that does not depend on the database size. We then train the efficient automatic re-ranking using the codes $I^i(\boldsymbol{x})$ and $\tilde{I}^i(\boldsymbol{x})$ together. This *pairwise additive decoding* ranker is then used at search time to find a second smaller shortlist $\mathcal{S}_{\mathrm{pairs}}$ during the search, see Figure 3. See App. B for an example.

## 4 EXPERIMENTAL VALIDATION

### 4.1 EXPERIMENTAL SETUP

**Datasets and metrics.** Following Huijben et al. (2024), we evaluate QINCo2 against previous baselines on the four datasets described in Table 1, spanning across various modalities, dimensions and train set sizes. We use the full training split during training, and use the database split to report the compression performance (MSE) on 1M vectors, and nearest-neighbor recall percentages at rank 1 (R@1) among 1M database vectors with 10k query vectors. We report recall at ranks 10 (R@10) and 100 (R@100) in Table S4 in the supplementary material; these metrics follow the same trends as those observed for R@1. Additionally, for Deep1B and BigANN, we use the 1B database for similarly search to evaluate IVF-QINCo2.

Table 1: **The datasets used in our experiments.**

| Dataset | Dim. | Train vecs. | Data type |
| --- | --- | --- | --- |
| Deep1B (Babenko & Lempitsky, 2016) | 96 | 358M | CNN image emb. |
| BigANN (Jégou et al., 2011) | 128 | 100M | SIFT descriptors |
| Facebook SimSearchNet++ (FB-ssnpp) (Simhadri et al., 2021) | 256 | 10M | SSCD image emb. |
| Contriever (Huijben et al., 2024) | 768 | 20M | Contriever text emb. |

**Architecture details.** Unless specified otherwise, we use the model architectures listed in Table 2. We use QINCo2-L for all the vector compression experiments (Section 4.2), as we want to see the impact of large models against the best results of other methods. The smaller models are used for search experiments, where

Table 2: **QINCo2 model architectures.**

| | # res. blocks ($L$) | emd. dim. ($d_{\mathrm{e}}$) | hid. dim. ($d_{\mathrm{h}}$) |
| --- | --- | --- | --- |
| QINCo2-S | 2 | 128 | 256 |
| QINCo2-M | 4 | 384 | 384 |
| QINCo2-L | 16 | 384 | 384 |

Table 3: **Comparison to state of the art methods for compression (MSE) and retrieval (R@1).** Ablation of model improvements w.r.t. QINCO are in *italics*, and the best results are in **bold**.

| | | BigANN1M | | Deep1M | | Contriever1M | | FB-ssnpp1M | | Train time |
|---|---|---|---|---|---|---|---|---|---|---|
| | | MSE $(\times 10^4)$ | R@1 | MSE | R@1 | MSE | R@1 | MSE $(\times 10^4)$ | R@1 | BigANN |
| **8 bytes** | OPQ | 2.95 | 21.9 | 0.26 | 15.9 | 1.87 | 8.0 | 9.52 | 2.5 | — |
| | RQ | 2.49 | 27.9 | 0.20 | 21.4 | 1.82 | 10.2 | 9.20 | 2.7 | — |
| | LSQ | 1.91 | 31.9 | 0.17 | 24.6 | 1.65 | 13.1 | 8.87 | 3.3 | — |
| | UNQ (Morozov & Babenko, 2019) | 1.51 | 34.6 | 0.16 | 26.7 | — | — | — | — | — |
| | QINCO (Huijben et al., 2024) | 1.12 | 45.2 | 0.12 | 36.3 | 1.40 | 20.7 | 8.67 | 3.6 | — |
| | QINCO (reproduction) | 1.13 | 45.3 | 0.12 | 35.6 | 1.40 | 20.6 | 8.66 | 3.8 | 127:38 |
| | *+ improved training* | *1.14* | *45.4* | *0.12* | *35.9* | *1.39* | *20.7* | *8.63* | *3.8* | *14:53* |
| | *+ improved architecture* | *1.09* | *45.9* | *0.12* | *36.7* | *1.38* | *21.2* | *8.63* | *4.0* | *21:42* |
| | *+ candidates pre-selection* | *1.10* | *45.5* | *0.12* | *36.7* | *1.38* | *20.6* | *8.64* | *3.9* | *8:23* |
| | *+ beam-search* | *0.85* | *50.6* | *0.10* | *43.9* | *1.34* | *22.7* | *8.18* | *4.4* | *57:39* |
| | **+ evaluate with larger beam (QINCO2)** | **0.82** | **52.3** | **0.09** | **45.1** | **1.34** | **23.1** | **8.14** | **4.5** | — |
| **16 bytes** | OPQ | 1.79 | 40.5 | 0.14 | 34.9 | 1.71 | 18.3 | 7.25 | 5.0 | — |
| | RQ | 1.30 | 49.0 | 0.10 | 43.0 | 1.65 | 20.2 | 7.01 | 5.4 | — |
| | LSQ | 0.98 | 51.1 | 0.09 | 42.3 | 1.35 | 25.6 | 6.63 | 6.2 | — |
| | UNQ (Morozov & Babenko, 2019) | 0.57 | 59.3 | 0.07 | 47.9 | — | — | — | — | — |
| | QINCO (Huijben et al., 2024) | 0.32 | 71.9 | 0.05 | 59.8 | 1.10 | 31.1 | 6.58 | 6.4 | — |
| | QINCO (reproduction) | 0.33 | 72.4 | 0.05 | 59.7 | 1.13 | 29.9 | 6.56 | 6.6 | 284:53 |
| | *+ improved training* | *0.33* | *72.4* | *0.05* | *59.5* | *1.09* | *31.4* | *6.54* | *6.8* | *37:50* |
| | *+ improved architecture* | *0.31* | *72.8* | *0.05* | *60.7* | *1.08* | *30.5* | *6.54* | *6.5* | *48:15* |
| | *+ candidates pre-selection* | *0.30* | *71.9* | *0.05* | *61.1* | *1.08* | *31.1* | *6.55* | *6.8* | *18:57* |
| | *+ beam-search* | *0.20* | *78.2* | *0.03* | *67.4* | *1.03* | *33.6* | *6.04* | *7.9* | *130:47* |
| | **+ evaluate with larger beam (QINCO2)** | **0.19** | **79.3** | **0.03** | 67.1 | **1.02** | **34.0** | **5.98** | **7.5** | — |

time efficiency matters as well. We set $A = 16, B = 32$ during training, and $A = 32, B = 64$ during evaluation for all models. When candidate pre-selection is used without beam search ($B = 1$), we use $A = 32$ during training. We fix the codebook size to $K = 256$, which results in a single byte encoding per step. Following prior work, see *e.g.* (Huijben et al., 2024; Morozov & Babenko, 2019), we consider 8-byte and 16-byte vector encodings in most of our experiments. We also experiment with 32-byte encoding for large-scale search.

**Baselines.** We compare QINCO2 against the results of OPQ (Ge et al., 2013), RQ (Chen et al., 2010), LSQ (Martinez et al., 2018), using the results reported by Huijben et al. (2024) which were obtained using the implementations in the Faiss library (Douze et al., 2024). We also compare against the neural quantization baselines QINCO (Huijben et al., 2024) and UNQ (Morozov & Babenko, 2019), citing the results reported in the original papers. As we build upon QINCO, we also report scores of our reproduction of this quantizer.

## 4.2 VECTOR COMPRESSION

**Main results.** In Tab. 3 we evaluate how QINCO2 improves over QINCO, by gradually introducing the changes to our reproduction of QINCO, and also compare to other state-of-the-art results. We report the MSE and R@1 metrics, as well as the training time for the BigANN1M models. Given the same training procedure, the training time is roughly proportional to the encoding time.

In general, our reproduction of QINCO closely follows the results reported by Huijben et al. (2024). When using our improved training recipe (*+improved training*), we observe a more than seven-fold reduction in training time w.r.t. QINCO (reproduction), while obtaining similar MSE. When adding our improved architecture (*+improved architecture*), we observed improved MSE and R@1 values across all datasets at both 8-byte and 16-byte compression (except for R@1 at 16 bytes on Contriever), at the cost of increased training time (but still remaining much faster than the QINCO reproduction). Adding candidate pre-selection ($A = 32, B = 1$) leads to a small degradation in MSE and R@1 in some cases, but makes training more than 2.5 times faster. Enabling beam search ($A = 16, B = 32$) leads to a substantial improvement in MSE and R@1. Although this has a substantial impact on the training time, training remains more than two times faster compared to QINCO (reproduction). Finally, we use a larger beam size for vector encoding at evaluation ($A = 32, B = 64$) than the one used for training ($A = 16, B = 32$). While not impacting the training time, the larger beam consistently improves the MSE across datasets and bitrates.

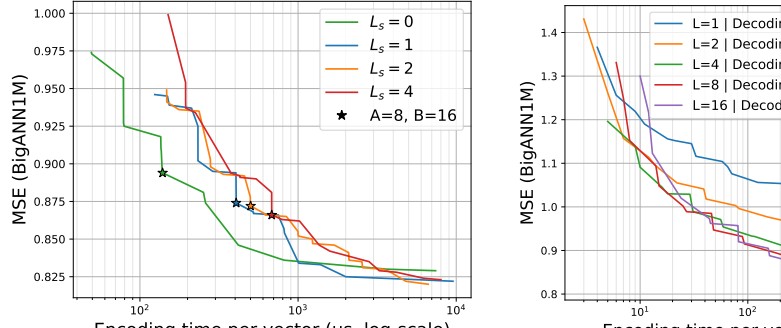

Figure 4: **Pareto fronts of quantization error on the BigANN1M dataset using 8 bytes codes.**
*Left:* Models with $L = 8$ blocks, and each curve using a different number of blocks $L_s$ for pre-selection. Each curve covers models that differ in the number of pre-selected codewords $A$ and beam size $B$. Models are trained using $A = 8$ and $B \in \{2, \ldots, 32\}$, and evaluated with $A \in \{8, \ldots, 64\}$, and $B \in \{2, \ldots, 128\}$. Stars show models using $A = 8, B = 16$ for evaluation. All models have the same decoding speed. *Right:* models with different number of residual blocks $L$. For models on one curve, the decoding time is fixed, and the encoding time is varied by changing $A$ and $B$.

Overall, we obtain results that substantially improve over earlier state-of-the-art results, reducing both the MSE and R@1 metric on all four datasets. For example, compared to UNQ (Morozov & Babenko, 2019) on the Deep1M dataset, we reduce the MSE more than two-fold from 0.07 to 0.03 for 16-byte codes, and improve the R@1 from 26.7% to 45.1% for 8-byte codes. Compared to QINCo on the BigANN dataset, we reduce MSE from 0.32 to 0.19 for 16-byte codes, and improve nearest-neighbor search accuracy from 45.2% to 52.3% and for 8-byte codes.

**Analysis of codeword pre-selection.** Figure 4 (left) shows the impact of the candidate pre-selection model depth $L_s$ on the MSE as a function of the encoding time, for fixed decoding time. For a given number of pre-selected candidates $A$ and beam size $B$, a deeper pre-slection model (higher $L_s$) reduces the MSE at the expense of increased encoding time. However, when varying $A, B$ settings, the models with $L_s = 0$ blocks —which just perform pre-selection based on a learned codebook— are Pareto-optimal for all settings with encoding times under 1 ms, thanks to the faster pre-selection of candidates. We therefore use this setting of $L_s = 0$ in all other experiments, relying on encoding parameters $A, B$ to set speed-accuracy tradeoffs.

**Trade-offs between encoding and decoding time.** In Figure 4 (right) we consider the MSE quantization error induced by models with different decoding speeds. For each model, we vary the encoding speed using different $A, B$ settings. The quantization error can be reduced with models that are deeper (higher decoding time), and using more exhaustive search for encoding (more pre-selected codewords and larger beam). However, for a given MSE the decoding time can be significantly reduced when compensated by higher encoding time. For example, an MSE of 1.0 can be obtained by a model with $L = 16$ residual blocks —decoding $0.27 \mu s$, encoding $20 \mu s$— or by a $L = 1$ model which decodes $3\times$ faster at but encodes $50\times$ slower —decoding $0.08 \mu s$, encoding 1 ms. These results show that QINCo2 allows trading compute between encoding and decoding, motivating the use of smaller models for large-scale retrieval experiments, where decoding time is crucial.

**Architecture sweep.** To obtain insight into the most effective hyperparameters settings, we present the results of a joint sweep over the number of residual blocks ($L$), the embedding and hidden dimensions ($d_e, d_h$) and the encoding parameters ($A, B$) in Figure 5, both for training and evaluation, using 8-bytes codes on BigANN. We also include several operating points of QINCo for reference. First, we notice that for a given MSE level, QINCo2 reduces the encoding time by about one order of magnitude compared to QINCo. Additionally, our QINCo2-S model used with $A = B = 8$ yields substantial improvements both in MSE and speed compared to all QINCo variants. For the operating points on the Pareto front, we notice a certain degree of correlation between the encoding speed (marker shape) and decoding speed (marker color) of the points on the Pareto front, with both increasing while MSE is improved. Note that this is in line with the Pareto front observed in Figure 4 (right). The most expensive encoding settings are mostly used in combination with the heaviest networks, yielding the best MSE results. When traversing the front left-to-right, at first shallow models are more optimal (with depth $L$=1 or 2), and MSE is improved by expanding the encoding search. Then, the Pareto front enters a phase where $L = 4$ models are optimal and the

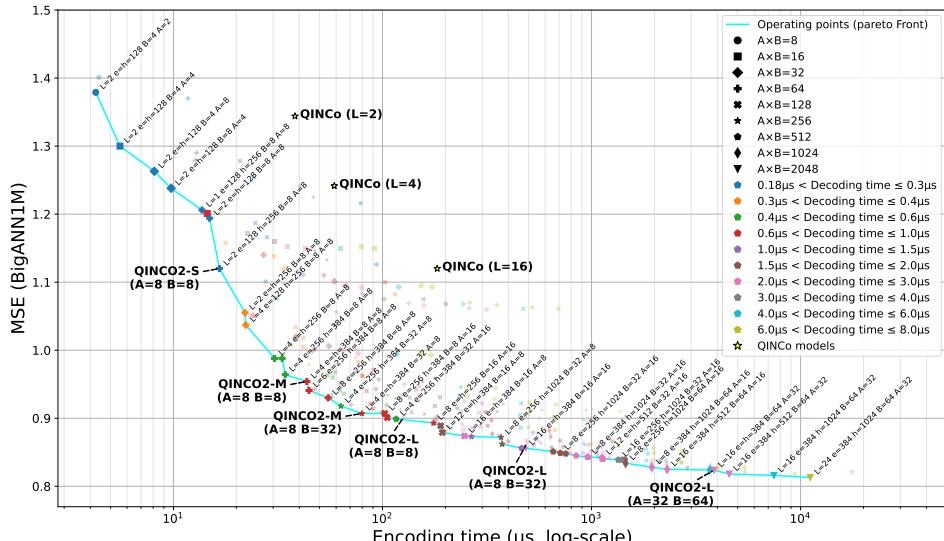

Figure 5: **Pareto-optimal front of QINCo2 operating points for MSE and encoding time.**
Evaluation on 10M vectors, varying the models parameters $A$, $B$, $L$, $d_{\mathrm{e}}$ and $d_{\mathrm{h}}$ when encoding using $M = 8$ bytes on BigANN1M. Models are trained with different $A, B \in \{16, 32\}$, and evaluated with a range of values up to $A = 64$ and $B = 32$. Marker shape is set according to the product of the encoding parameters $A \times B$, and color according to the decoding time, determined by the network depth ($L$) and width ($d_{\mathrm{e}}, d_{\mathrm{h}}$). Results for QINCo are shown as yellow stars for comparison.

|  |  | BigANN1M | | Deep1M | |
|---|---|---|---|---|---|
|  |  | R@1 | $n_{short}=10$ | R@1 | $n_{short}=10$ |
| **8 bytes** | QINCo2-S (no shortlist) | **47.9** | — | **38.1** | — |
|  | AQ | 12.3 | 31.5 | 10.2 | 26.0 |
|  | RQ | 12.1 | 31.3 | 10.2 | 25.6 |
|  | RQ, w/ $\frac{M}{2} = 4$ consecutive code-pairs | 17.8 | 39.8 | 16.4 | 32.0 |
|  | RQ, w/ $2M = 16$ optimized code-pairs | 28.2 | **46.2** | 24.0 | **36.6** |
| **16 bytes** | QINCo2-S (no shortlist) | **73.2** | — | **63.1** | — |
|  | AQ | 16.6 | 45.2 | 16.4 | 42.4 |
|  | RQ | 16.0 | 44.4 | 15.5 | 41.7 |
|  | RQ, w/ $\frac{M}{2} = 8$ consecutive code-pairs | 21.5 | 54.7 | 20.6 | 49.5 |
|  | RQ w/ $2M = 32$ optimized code-pairs | 35.0 | **66.9** | 33.4 | **58.6** |

Table 4: **Search results using QINCo2 decoder and approximate decoders for QINCo2 codes.** For each combination of dataset and bitrate, we report the retrieval accuracy over 1M vectors, as well as the accuracy of QINCo2-S over a shortlist of 10 elements generated by the method.

search span increases from $A \times B = 16$ to 512. Finally, the lowest MSE and highest encoding times are obtained with depths ranging from 8 to 24, with search spans ultimately reaching $A \times B = 2,048$.

## 4.3 LARGE-SCALE VECTOR SEARCH

**Approximate decoders.** In Table 4 we consider a preliminary experiment comparing a small QINCo2 model with fast lookup-based decoders trained on fixed QINCo2 codes, including AQ, RQ and our pairwise additive decoders. The AQ decoder is learned by solving a single large least-squares problem, while the other ones are learned by solving up to $2M$ successive smaller least-squares problems. We consider the direct recall of these decoders, as well as the recall when using them to establish a shortlist of ten elements which is then re-ranked with QINCo2. We observe that the RQ decoder yields only a small degradation w.r.t. the AQ which is more expensive to train. When comparing R@1, both of these methods perform much worse though than the QINCo2 decoder, losing more than 73% of its search accuracy. However, accuracy is more than doubled when using a pairwise additive decoder with $2M$ pairs. When used to form a shortlist of ten elements, single-code approximate methods still lag behind QINCo2 (12 points of R@1 for 8-bytes codes, more than 20 points for 16-bytes). But decoding using optimized code-pairs reduces the gap considerably (at most 1.7 points for 8-bytes and 6.3 points for 16-bytes), with minimal computational overhead: ten calls to $F_{\mathrm{QI}}$, compared to a million when using the QINCo2 decoder without shortlists.

**Large-scale search efficiency.** In Figure 6 we plot the search accuracy as a function of speed, reported as queries per second on a single CPU, when searching over 1B database vectors from BigANN using 8, 16 and 32 byte representations. In Figure S2 in the supplementary material we

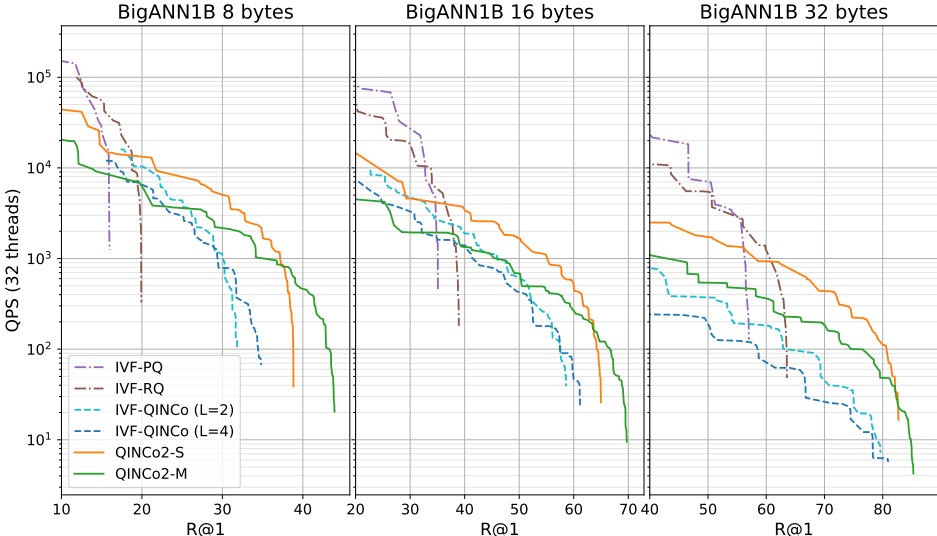

Figure 6: Retrieval accuracy/efficiency trade-off on the Bigann1B dataset in terms of queries per second (QPS) and recall (R@1) when combining PQ, RQ, QINCo, and QINCo2 with IVF.

provide similar results when using the Deep1B dataset. We compare to PQ and RQ (with beam size $B = 20$) baselines, using their implementation in the Faiss library, as well as QINCo models with $L = 2$ and $L = 4$ residual blocks, as used in the experiments of Huijben et al. (2024). For QINCo2 we consider two models: QINCo2-S has a depth and width similar to the QINCo ($L = 2$) model, striking a good balance between speed and accuracy for most settings, whereas QINCo2-M is slightly larger than the second QINCo model, reaching the highest overall retrieval accuracy. We plot Pareto fronts for all compared methods by changing the hyperparameters of the IVF search: the number of IVF buckets that are accessed, the shortlist size(s), and the efSearch parameter that is used in the HNSW algorithm to find the IVF centroids closest to the query.

We find that at the highest search speeds, the PQ and RQ baselines yield the best results, but also that their accuracy quickly saturates at low recall values when more compute is used. With QINCo2, we are able to attain significantly higher recall values: adding more than 20 recall points for each bitrate in the high-compute regime. The search speed at which QINCo2 starts to yield more accurate results than the PQ and RQ baselines is generally situated in the range 1,000 to 10,000 queries per second. Compared to the two QINCo models, our QINCo2-S consistently yields higher search accuracy across the full range of search speeds. Among our two QINCo2 models, QINCo2-M is generally slower than QINCo2-S, but attains the highest accuracy in the high-compute regime.

## 5 CONCLUSION

We presented QINCo2, a neural residual quantizer in which codebooks are obtained as the output of a neural network conditioned on the vector reconstruction from previous steps. Our approach improves over QINCo in several ways. First, the training procedure and network architecture are optimized, which leads to similar results but reduces training time roughly by a factor six. Second, we introduce a candidate pre-selection approach to determine a subset of the codewords for which we evaluate the QINCo2 network, further reducing encoding and training time roughly by a factor three. Third, given the accelerations, we use beam search for encoding and find it can greatly reduce the quantization error. Finally, for application of QINCo2 to billion-scale nearest neighbor search, we introduce pairwise look-up based decoders to obtain a fast approximate decoding of the QINCo2 codes, that are much more accurate than the AQ decoder used in QINCo. We conduct experiments on four different datasets, and evaluate performance in terms of MSE reconstruction and the search speed-accuracy trade-off, using different bitrates for both tasks. On both tasks and all datasets, QINCo2 consistently improves over QINCo and other baselines. In particular, we reduce MSE by 34% for 16-byte compression of BigANN, and push the maximum accuracy for billion-scale nearest neighbor search of the RQ baseline from under 40% to over 70% using 16-byte codes for BigANN. Beyond our contributions to implicit neural quantization, we believe it is interesting to explore the potential of our pairwise lookup-based additive decoder for other quantizers in future work.

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

# A  IMPLEMENTATION DETAILS

## A.1  QINCO2 ARCHITECTURE

The architecture of the neural network $f_\theta$, illustrated in Figure 1, is formally described by Eqs. (10) to (13). This network is parameterized by a number of blocks $L$, and dimensions $d_e$ and $d_h$. At step $m$, a codeword $c \in \mathbb{R}^d$ is projected into an embedding space of dimension $d_e$ (Eq. (10)). Is it then conditioned on the previous reconstruction $\hat{x}^{m-1}$ by concatenating both variables (Eq. (11)), adding a residual connection after the projection. The output is then computed by a sequence of 2-layers residual MLPs (Eq. (12)). A residual connection is added at the end (Eq. (13)) to ensure stability.

In the following equations we denote by $\mathtt{L}_{d_1}^{d_2}$ a linear layer from dimension $d_1$ to $d_2$, and $\mathtt{P}_{d_1}^{d_2}$ a projection from $d_1$ to $d_2$ which is equal to $\mathtt{L}_{d_1}^{d_2}$ if $d_1 \neq d_2$, and to the identity function otherwise. These layers are implicitly parameterized by $\theta$. We then express $f_\theta(c^m|\hat{x}^m)$ as follows:

$$c_{\text{emb}} = \mathtt{P}_d^{d_e}(c^m), \tag{10}$$

$$v_0 = c_{\text{emb}} + \mathtt{L}_{d+d_e}^{d_e}\left(\text{Concat}[c_{\text{emb}}; \hat{x}^{m-1}]\right) \tag{11}$$

$$v_i = v_{i-1} + \mathtt{L}_{d_h}^{d_e}(\text{ReLU}(\mathtt{L}_{d_e}^{d_h}(v_{i-1}))) \tag{12}$$

$$f_\theta(\hat{x}^m, c^m) = c^m + \mathtt{P}_{d_e}^d(v_L) \tag{13}$$

The $\mathtt{L}_{d+d_e}^{d_e}$ layer (Eq. (11)) uses a bias term, all the others don't.

The number of parameters we use for our models listed in Table 2 increase the number of trainable parameters of up to a factor of 4.5 compared to QINCo. We include a comparison of the number of parameters between RQ, QINCo and QINCo2 in Table S1. As shown in Table 3, the increase in model parameters contributes to an improvement in MSE (see line *improved architecture*).

Table S1: **Number of parameters of RQ, QINCo and QINCo2 models on BigANN1M.**

| Model | RQ | QINCo (L=2) | QINCo (L=4) | QINCo (L=16) | QINCo2-S | QINCo2-M | QINCo2-L |
|---|---|---|---|---|---|---|---|
| **Parameters** | 0.26M | 1.4M | 2.3M | 7.8M | 1.6M | 10.8M | 35.6M |

## A.2 TRAINING QINCo2

We follow the same optimization approach as QINCo (Huijben et al., 2024). Using the loss function $\mathcal{L}(\boldsymbol{x}, \boldsymbol{q}) = \|\boldsymbol{x} - \boldsymbol{q}\|_2^2$, training QINCo2 amounts to finding the optimal parameters $\theta, C^1, \ldots, C^M$ for the quantization process Eq. (1). This defines the following optimization problem:

$$\underset{\theta, \boldsymbol{c}^1 \in C^1, \ldots, \boldsymbol{c}^M \in C^M}{\arg\min} \mathbb{E}_{\boldsymbol{x}} \left[ \underset{\boldsymbol{c}^1 \in C^1, \ldots, \boldsymbol{c}^M \in C^M}{\arg\min} \mathcal{L}\left(\boldsymbol{x}, F_\theta\left(\boldsymbol{c}^1, \ldots, \boldsymbol{c}^M\right)\right) \right], \tag{14}$$

where $F_\theta$ is a sequence of $M$ QINCo2 models, as defined by Eq. (4). We follow the optimization approach from Huijben et al. (2024) to solve Eq. (14) by alternating between solving the external and inner problems. For each batch, we first solve the inner problem by using our quantization process $\mathcal{Q}_{\text{QI-B}}$ (instead of $\mathcal{Q}_{\text{RQ}}$ for QINCo ; see Section 3.2). We then use a gradient-based method (AdamW, instead of Adam for QINCo, see below) to optimize parameters $\theta$ and $C^1, \ldots, C^M$.

Compared to QINCo, we change the training procedure in the following ways:

- Like QINCo we loop over 10M samples per epoch, but with a different segment of 10M vectors at each epoch to cover the full dataset. We set aside the same 10k vectors as QINCo for validation.

- We reduce the number of epochs to 70, while QINCo can train for hundred of epochs, relying on its scheduler for stopping.

- We normalize each dataset with a mean of 0 for each feature, and a standard deviation of 1 across all features, instead of normalizing to $[0; 1]$.

- We initialize the QINCo2 codebooks using noisy RQ codebooks. We train these codebooks for only 10 k-means iterations, for each of the $M$ codebook. We then add a Gaussian noise with standard deviation, $\sigma = s \times 0.025$ where $s$ is the per-feature standard deviation computed over the RQ codebooks.

- We initialize all the weights of the network using Kaiming uniform initialization, and initialize to zero all biases and weights of the down-projections $\mathsf{L}_{d_h}^{d_e}$ within the residual blocks.

- We use the AdamW optimizer (Loshchilov & Hutter, 2019) with default settings, except for a weight decay of $0.1$ (instead of using Adam).

- We use a gradient clipping set to $0.1$, and decrease it to $0.01$ on unstable experiments.

- We use a maximum learning rate of $0.0008$, and decrease it to $0.0001$ on unstable experiments (compared to $0.001$ for the base learning rate of QINCo).

- We use a cosine scheduler, with a minimum learning rate of $10^{-3}$ times the maximum learning rate (instead of reducing the learning rate on plateaus only).

- We increase the batch size to 1,024 on each of the 8 GPUs, for an effective batch size of 8,192 (compared to an effective batch size of 1024 for QINCo).

- At the end of each epoch, we reset every codeword that has not been used at all. We reset a codeword from the codebook $m$ using a uniform distribution with the same mean and standard deviation as the residuals quantized by step $m$, *i.e.* $\mu$ and $\sigma$ from the distribution $\boldsymbol{x} - \hat{\boldsymbol{x}}^{m-1}$. This is similar to the approach of Zheng & Vedaldi (2023) to re-initialize "dead" codewords.

Each of our choices increased the final results in early experiments, except for the reduced number of epochs, which we set to favor training speed. We show the effect of the revised training procedure in our experiments in the following section.

Table S2: Encoding and decoding complexity per vector in FLOPS in "big-O" notation, and indicative timings on BigANN1M using 32 CPU cores (in $\mu$s) with parameters: $D$=128; QINCo: $L$=2, $M$=8, $h$=256; UNQ: $h'$=1024; $b$=256; RQ: beam size $B$=5; QINCo2-S: $A$=8, $B$=32, $d_e = 128, d_h = 256$. In practice, at search time for OPQ and RQ rather than decoding we perform distance computations in the compressed domain, which takes $M$ FLOPS (0.16 ns).

| | Encoding | | Decoding | |
|---|---|---|---|---|
| | FLOPS | time | FLOPS | time |
| OPQ | $d^2 + Kd$ | 1.5 | $d(d+1)$ | 1.0 |
| RQ | $KMdB$ | 8.3 | $Md$ | 1.3 |
| UNQ | $h'(d + h' + Mb + MK)$ | 18.8 | $h'(b + h' + d + M)$ | 13.0 |
| QINCo | $KMd(d + Lh)$ | 823.4 | $Md(d + Lh)$ | 8.3 |
| QINCo2-S | $ABMd_e(d + Ld_h) + BKd$ | 2910.7 | $Md_e(d + Ld_h)$ | 6.2 |

To train efficiently, we encode each batch without maintaining activations for gradient computation, and then perform another forward-backward pass through $f_\theta$ using the codes selected by the encoding process. This is much faster as in this manner, the forward-backward pass is only executed for a single codeword per vector per step, rather than for all codebook elements.

### A.3 LARGE-SCALE SEARCH WITH FAISS

To implement IVF search with the AQ and RQ baselines we use the Faiss library (Douze et al., 2024).[1] The index factory names used for our experiments are `IVF1048576_HNSW32,RQ<BB>x8_Nqint8`, where `<BB>` is replaced by 8, 16 or 32, depending on the number of bytes used per vector. It indicates the use of $K_{IVF} = 2^{20}$ centroids, indexed with a HNSW graph-based index (32 links per node), followed by RQ filtering using `<BB>` bytes. This structure is used to perform the IVF-RQ search, where the codebooks are computed by Faiss directly. For IVF-QINCo2, we compute the IVF centroids and AQ codebooks and codes on GPU before filling this structure with them, as the RQ and AQ decoding function are the same. During evaluation of large-scale search with IVF-QINCo2, we use this structure to retrieve the $\mathcal{S}_{AQ}$ shortlist, which is re-ranked using pytorch implementations of pairwise additive decoding, and then QINCo2.

## B ADDITIONAL RESULTS

**Encoding and decoding complexity.** In Table S2 we compare QINCo2 to QINCo, UNQ, RQ and PQ in terms of complexity (FLOPS) and timings obtained on CPU. As our method QINCo2 has a variable encoding time depending on runtime parameters, we select values such that $A \times B = K = 256$ for fair comparison with QINCo.

**Bitrate reduction.** In Figure S1, we show how QINCo2 contributes to reducing the bitrate at a given reconstruction error compared to previous methods. Relative improvements in bitrate increase for lower MSE and higher number of quantization steps. For 8-bytes codes from previous methods, QINCo2 achieves similar error using at least 29% fewer bytes, but for 16-bytes codes, it achieves at least 54% reduction in bitrate.

---

[1]Acessed from https://github.com/facebookresearch/faiss.

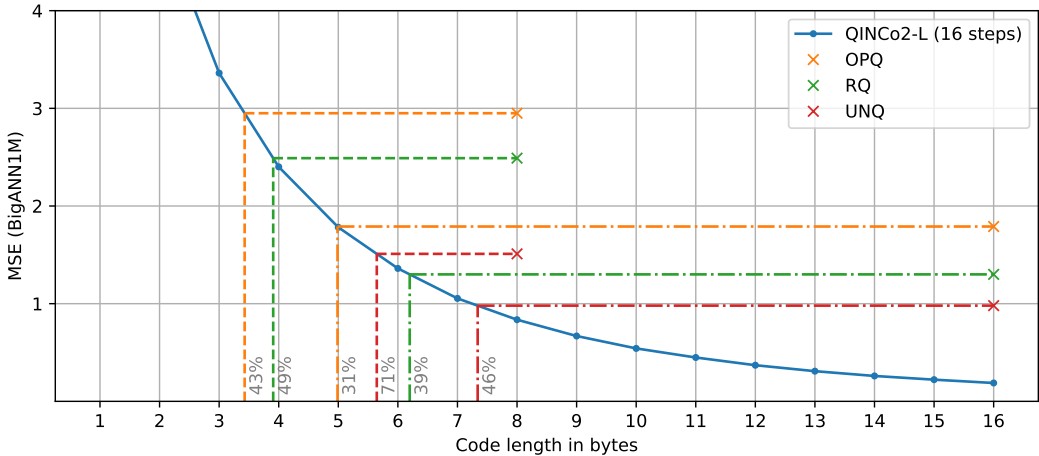

Figure S1: MSE of QINCO2 and previous methods at different bitrates. Dotted lines show the bitrate reduction of QINCO2 compared to previous methods at a fixed MSE.

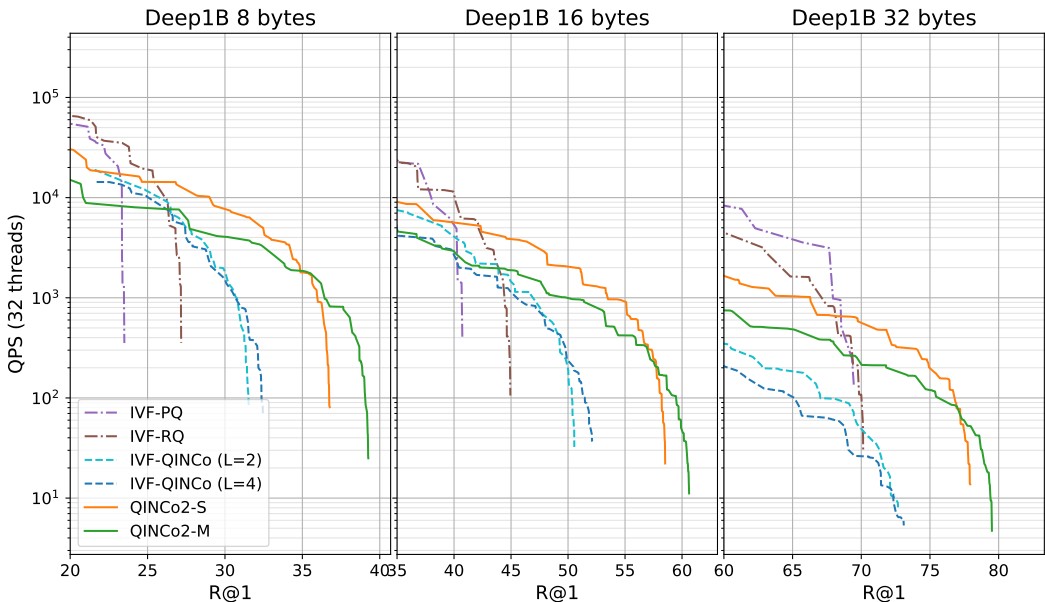

Figure S2: Search accuracy/efficiency trade-off on the Deep1B dataset in terms of queries per second (QPS) and recall (R@1) for PQ, RQ, QINCO and QINCO2 combined with IVF.

**Large-scale search.** In Figure S2 we report large-scale search performance on the Deep1B dataset. Overall we find the same trends as those observed for BigANN in Figure 6 in the main paper. In particular QINCO2-S strictly dominates QINCO for all operating points, and QINCO2 is able to attain significantly higher recall levels than the PQ, RQ and QINCO baselines.

**Dynamic rates.** In Figure S3 we compare QINCO2 models trained for different numbers of steps $M$, from 4 up to 16. We evaluate the MSE of these models after $m = 1, \ldots, 16$ steps. We find that for a given $m$ the MSE of all models trained for $M \geq m$ are nearly identical. This is in line with the findings of Huijben et al. (2024), and makes that QINCO2 trained with large $M$ can be used as a multi-rate codec as its performance is (near) optimal for usage with smaller codes.

**Changing beam size at evaluation.** Besides the model size, QINCO2 introduces two hyperparameters to control the accuracy: the number of pre-selected candidate codewords $A$ and the beam size

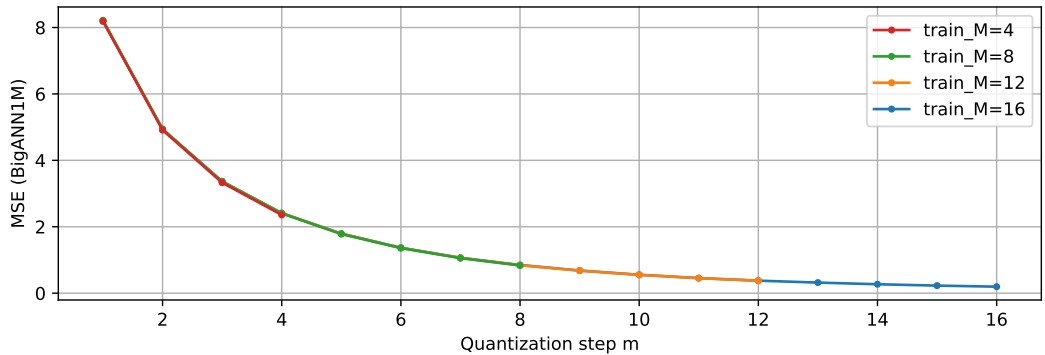

Figure S3: MSE for QINCo2 after quantization step $m$, for models trained with different $M$ values. All models use $L = 8$, $d_e = 384$, $d_h = 384$, $A = 16$ and $B = 32$.

Table S3: Pairs generated by our pairwise additive decoder on Deep1M for 8-bytes quantization, with corresponding MSE at each step. $i$ indicates a QINCo code ($I^i$), $\tilde{i}$ indicates a code extracted from the IVF centroid ($\tilde{I}^i$)

| **Step** | AQ + IVF | 1 | 2 | 3 | 4 | 5 | 6 | 7 | 8 |
|---|---|---|---|---|---|---|---|---|---|
| **Pair** | - | $(1,2)$ | $(3,4)$ | $(5,6)$ | $(7,8)$ | $(1,\tilde{1})$ | $(2,\tilde{1})$ | $(3,\tilde{1})$ | $(4,\tilde{1})$ |
| **MSE** | 3.78 | 3.20 | 2.77 | 2.56 | 2.43 | 2.33 | 2.23 | 2.17 | 2.13 |
| **Step** | - | 9 | 10 | 11 | 12 | 13 | 14 | 15 | 16 |
| **Pair** | - | $(5,\tilde{1})$ | $(6,\tilde{1})$ | $(7,\tilde{1})$ | $(1,\tilde{2})$ | $(2,\tilde{2})$ | $(8,\tilde{2})$ | $(3,\tilde{2})$ | $(1,\tilde{3})$ |
| **MSE** | - | 2.10 | 2.08 | 2.06 | 2.04 | 2.03 | 2.02 | 2.01 | 2.01 |

$B$. The number of evaluations of the QINCo2 network is equal to the product $A \times B$, and the encoding time during and after training is roughly proportional to this quantity. Note, however, that these parameters can be set differently during training, and when using the model for encoding once the model has been fully trained.

In Figure S4 we report the impact on MSE of changing $A$ at evaluation time when using the same model trained with different $A$, when using a fixed beam size of $B = 16$. The results show that regardless of the $A$ used for training, the MSE saturates around $A = 24$ at evaluation, ensuring that $A = 32$ will yield close-to-optimal results for encoding. In order to obtain the best performance, $A = 16$ for training is a good choice: smaller values leads to worse MSE, while $A = 32$ does not lead to a noticeable improvement.

In Figure S5 we similarly consider the effect of changing the beam size $B$ during evaluation for models trained with different beam sizes. In this case, we observe that most models lead to similar MSE values for a given value of $B$ for evaluation. Only the model trained with $B = 2$ leads to significantly worse results when using large beams for evaluation, and similarly the model trained with $B = 32$ yields worse results than others when using relatively small beams for evaluation ($B \leq 16$). Overall, models keep improving the MSE when increasing the beam even up to size $B = 64$ for evaluation. Therefore, it seems a good strategy to train models with $B = 8$, and use larger $B$ for evaluation when more accuracy is required.

**Pairwise additive decoding with IVF example.** We show in Table S3 an example of code-pairs generated by our pairwise decoder (Section 3.3) on the Deep1M dataset using 8-bytes quantization. We additionally show the MSE of the reconstructed codebooks at each step of this decoder, starting from the reconstruction yielded by the IVF and AQ steps.

**Retrieval accuracy with relaxed settings.** We show in Table S4 both the exact retrieval accuracy (R@1), and the relaxed accuracies R@10 and R@100 where a request is correctly answered if the nearest neighbour is within the 10 or 100 first elements returned by the method. We note that the the ranking between methods stays the same for each metric, but the accuracy gaps are reduced as the metrics get less restrictive.

**Latency.** In Figure 6 and Figure S2, we show the speed of queries as the number of queries per second, with batched requests. In some settings, it might be more interesting to look as *latency* instead

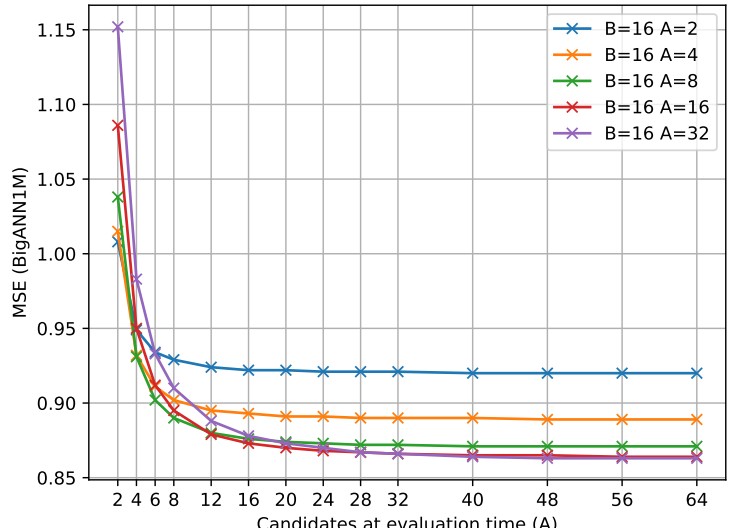

Figure S4: MSE of QINCo2-L models (with $L = 8$ and $B = 16$), trained with five different number of candidates $A$, when changing $A$ at inference time.

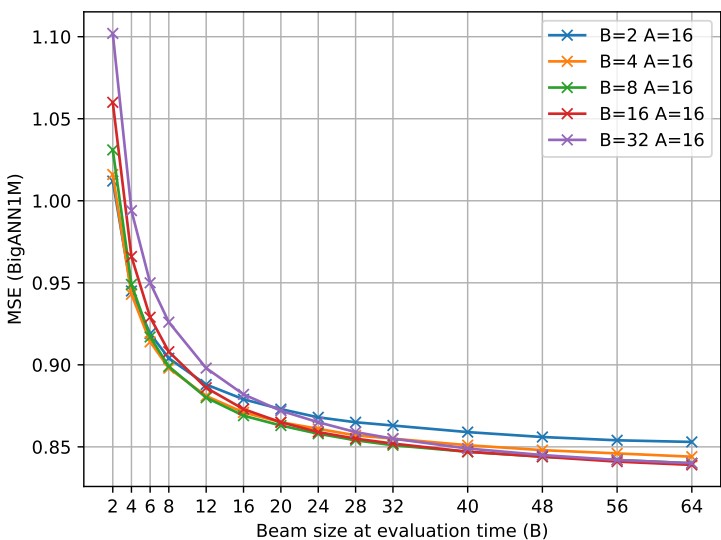

Figure S5: MSE of QINCo2-L models (with $L = 8$ and $A = 16$), trained with five different beam sizes $B$, when changing $B$ at inference time.

Table S4: Comparison to state of the art methods for retrieval (R@1, R@10 and R@100). The best results are in **bold**.

| | | BigANN1M | | | Deep1M | | | Contriever1M | | | FB-ssnpp1M | | |
|---|---|---|---|---|---|---|---|---|---|---|---|---|---|
| | | R@1 | R@10 | R@100 | R@1 | R@10 | R@100 | R@1 | R@10 | R@100 | R@1 | R@10 | R@100 |
| **8 bytes** | OPQ | 21.3 | 64.3 | 95.6 | 15.1 | 51.1 | 87.9 | 8.5 | 24.3 | 50.4 | 2.5 | 5.0 | 11.2 |
| | RQ | 27.9 | 75.2 | 98.0 | 21.9 | 64.0 | 95.2 | 9.7 | 27.1 | 52.6 | 2.7 | 5.9 | 14.3 |
| | LSQ | 30.6 | 78.7 | 98.9 | 24.5 | 68.8 | 96.7 | 13.1 | 34.9 | 62.5 | 3.5 | 8.0 | 18.2 |
| | UNQ | 39.7 | 88.3 | 99.6 | 29.2 | 77.5 | 98.8 | – | – | – | – | – | – |
| | QINCo | 45.2 | 91.2 | 99.7 | 36.3 | 84.6 | 99.4 | 20.7 | 47.4 | 74.6 | 3.6 | 8.9 | 20.6 |
| | QINCo2-L | **52.3** | **95.2** | **99.9** | **45.1** | **90.8** | **99.8** | **23.1** | **51.5** | **77.3** | **4.5** | **11.0** | **24.2** |
| **16 bytes** | OPQ | 41.3 | 89.3 | 99.9 | 34.7 | 81.6 | 98.8 | 18.1 | 18.1 | 65.8 | 5.2 | 12.2 | 27.5 |
| | RQ | 49.1 | 94.9 | 100.0 | 42.7 | 90.5 | 99.9 | 19.7 | 43.8 | 68.6 | 5.1 | 12.9 | 30.2 |
| | LSQ | 49.8 | 95.3 | 100.0 | 41.4 | 89.3 | 99.8 | 25.8 | 55.0 | 80.1 | 6.3 | 16.2 | 35.0 |
| | UNQ | 64.3 | 98.8 | 100.0 | 51.5 | 95.8 | 100.0 | – | – | – | – | – | – |
| | QINCo | 71.9 | 99.6 | 100.0 | 59.8 | 98.0 | 100.0 | 31.1 | 62.0 | 85.9 | 6.4 | 16.8 | 35.5 |
| | QINCo2-L | **79.3** | **99.9** | **100.0** | **67.1** | **99.2** | **100.0** | **34.0** | **66.5** | **89.4** | **7.5** | **19.6** | **40.7** |

of *queries per second*. To this end, we compute the **latency** of a single query for two operating points on the 16-bytes and 32-bytes BigANN1B large-scale search curves for IVF-QINCO2 and IVF-RQ. To ensure a fair comparison, we selected points close to both pareto-optimal fronts, where IVF-QINCO2 and IVF-RQ have approximately the same QPS & R@1. We use a single CPU for timing. On BigANN1B (16 bytes) at a point with R@1=37 and QPS=2700, RQ has a latency of 10.78ms, and 9.10ms for QINCO2. On BigANN1B (32 bytes) at a point with R@1=62 and QPS=350, RQ has a latency of 71.54ms, and 22.25ms for QINCO2. While RQ uses larger parameters for the *faiss* query to achieve this accuracy, QINCo2 uses a less accurate *faiss* search combined with a precise re-ranking. The smaller latency for QINCo2 might indicate that the *faiss* search benefits more from batched queries, and that QINCo2 can bring substantial improvements to speed with a small number of queries.

