# OpenReview forum: "Qinco2: Vector Compression and Search with Improved  Implicit Neural Codebooks"
_ICLR.cc/2025/Conference — ICLR 2025 Poster_

### Official Review · Reviewer_tRnB · 2024-10-31

**Soundness:** 3
**Presentation:** 3
**Contribution:** 3
**Rating:** 8
**Confidence:** 4

**Summary:**

This paper enhances QINCo in both the encoding and decoding processes. To tackle the significant complexity of encoding, the authors introduce codeword pre-selection and beam search strategies, which improve encoding efficiency and approximation capabilities. Additionally, to mitigate the limited search accuracy of the AQ decoder, the authors propose a fast approximate decoder based on pairwise additive code, which creates accurate shortlists for fast searching. Experimental results demonstrate that QINCo2 improves both efficiency and search accuracy.

**Strengths:**

1.	The proposed method seems concise and effective, especially in speeding-up the QINCo encoding and searching process.
2.	The pairwise additive decoding looks like an effective tool to create more accurate approximation of non-independent neural codebooks.
3.	The experiments and analysis are quite extensive and the improvements are significant.
4.	The paper is well-written and easy to read.

**Weaknesses:**

1.	In Table 3, “Improved Architecture” slightly improves the search accuracy on BigANN and Deep datasets with lower vector dimension. Since the performance of original QINCo is largely affected by the network scale, the question is whether the “Improved Architecture” in QINCo2 affects the performance by improving the network parameters. It is better to provide the comparison of parameters.
2.	Compared to the original QINCo, the “Improved Training” approach used in this paper incorporates more training samples. Results in Table 3 shows that the introduction of large training set brings limited performance improvement. With a fixed training epoch of 70 and the sequential acquisition of each 10M splits, wonder if the model achieves optimal convergence with such a large training set.

**Questions:**

1.	The dataset names in Table 3 should be consistent with other results in Sec. 4.2, i.e., BigANN1M, Deep1M, Contriever1M, and FB-ssnpp1M.
2.	A little confused on the “2M successive least-squares problems” in RQ-based codebook approximation (mentioned in Sec. 4.3), as there are only M steps in RQ.
3.	The R@10 and R@100 results of QINCo2 are not included in this paper, despite the authors' claim in Section 4.1 that recall percentages at ranks 1, 10, and 100 have all been considered.

---

> ### Author Response · Authors · 2024-11-21
>
> We’re happy to read that the reviewer found our paper “well-written and easy to read” and that
> “experiments and analysis are quite extensive and the improvements are significant.” We address comments in the weaknesses and questions section below.
>
> **W1 (“comparison of parameters”)** The increased trainable parameter count indeed contributes to the improved MSE (“improved architecture“ in Table 3). We added a comparison of the parameter count in the first section of the appendix, and thank the reviewer for the suggestion. QINCo2-L uses 35.6M parameters, compared to 7.8M for the larger QINCo1. For the efficient large-scale search setting the difference is much smaller: QINCo2-S uses 1.6M parameters where QINCo1 (L=2) uses 1.4M.
>
> **W2 (“optimal convergence”)** Our training procedure was designed to reduce training cost while preserving or improving the MSE compared to QINCo1. We can indeed reach lower error by increasing the training time. Following the reviewer suggestion, we ran experiments on BigANN1M and Deep1M (8 and 16 bytes) with more epochs, and observed an improvement of 0.5% up to 1% in MSE, while roughly doubling the training time. This indicates that our models are close to optimal convergence, and could have slightly improved results for a high additional training cost, which we did not deem worth for training large models compared to other improvements we studied in our paper.
>
> **Q1 (“consistent names”)** We thank the reviewer for catching this, and we updated the names in the paper.
>
> **Q2 (“2M successive least-squares problems”)**  Some of the methods referred in Sec 4.3 and in Table 4 learn new codebooks based on combinations of two codes, as explained in the paragraph “Pairwise additive decoding” in Section 3.3. We take fixed codes i and j each indexing a vocabulary of K codewords, we combine these to create a new code indexing over a vocabulary of K^2 elements. As we have M codes, it means a maximum of M*(M-1) pairwise combinations. In our experiments we consider using pu to a maximum of 2M of such pairwise codebooks.
>
> **Q3 (“R@10 and R@100 scores”)** Thank you for noticing this. In general we notice that the ordering of methods in these more relaxed metrics is similar, while the differences between them are reduced. This can also be seen in Table S2 of (Huijben et al., 2024). For sake of completeness, we added Table S4 to the appendix where we report the R@10/100 metrics for the experiments considered in Table 3 of the main paper.

---

> > ### Comment · Reviewer_tRnB · 2024-11-25
> >
> > Thanks Authors for the further explanation! I'm good with the answers.

---

### Official Review · Reviewer_xDMj · 2024-11-03

**Soundness:** 2
**Presentation:** 3
**Contribution:** 2
**Rating:** 6
**Confidence:** 2

**Summary:**

QINCO2 is an improved version of the original QINCO model for residual MCQ. It improves search efficiency in large datasets and reconstruction error. Both methods use neural network to dynamically adapt codebooks after each step of residual quantization. Instead of static codebook (conventional RQ), QINCO2 (and QINCO) uses neural network to adjust the codebook based on the current approximation and base codebook values. The network inputs the residual vector and partial reconstruction and produces centroids that more accurately encode the residuals. The original QINCO dramatically increased computational complexity of the quantization process and memory usage.
QINCO2 improves encoding speed by introducing codeword pre-selection which narrows down the search of centroids. It uses another neural network of smaller parameters to calculate top $A$ candidates (among possible centroids) which is further used for adaptive quantization. Furthermore, QINCO2 applies beam search to improve quantization quality by exploring multiple encoding paths in parallel, which helps to minimize the quantization error and refine the encoded representation more accurately.
To address the high computational cost during decoding, QINCO2 introduces a pairwise additive decoder, which enables faster approximate decoding by combining pairs of codewords, effectively capturing dependencies between codewords

**Strengths:**

- Proposed method significantly improves quantization error and retrieval accuracy
- It is faster for retrieval tasks, which is important for industry scale applications

**Weaknesses:**

The theoretical contribution is rather low. Authors mainly engineered existing methods together to improve inference of the model.
The paper is very hard to follow, it is not completely clear why introducing another neural network for pre-selection can speed it up (furthermore, increasing training training time)

**Questions:**

N/A

---

> ### Author Response · Authors · 2024-11-21
>
> We would like to thank the reviewer for acknowledging that our work “significantly improves quantization error and retrieval accuracy”. We address the points raised in the review point-by-point below.
>
> **W1 (“theoretical contribution”)** We do not claim theory contributions in our paper; our contributions are methodological (pre-selection, pairwise decoding) and technical (beam search, improved architecture and training) in nature. Our contributions lead to significant advances in reconstruction and search accuracy, as demonstrated through extensive experimental results.
>
> **W2 (“paper hard to follow”)** We’re sorry that some parts of the paper might have been less clear. If the reviewer can point to specific passages we’re happy to clarify those further.
>
> **W3 (pre-selection not clear)** During the QINCo1 encoding process, all codewords from each codebook are forwarded through the neural network to obtain their adaptation and select the more accurate one, which significantly improves quantization accuracy.
> Codeword pre-selection, one of our main contributions, allows to forward only a subset of the codewords through the (computationally expensive) neural network. For this to be effective, the key is to pre-select the codewords by a technique that is more computationally efficient. We found that pre-selection using L2 distance to a learned codebook (which we compare to also using smaller networks) leads to important performance gains during the encoding process.
> As the encoding process (forward pass) is also the bottleneck during training, our method reduces training time, even if it has slightly more parameters to learn.

---

> > ### Comment · Reviewer_xDMj · 2024-11-26
> >
> > I have carefully examined the responses, other reviews, and the paper itself. Both this paper and the original Qinco paper leave the optimization process of the model somewhat unclear. While the presented loss function is non-differentiable, the authors state that SGD is employed, yet they provide only vague details about how the optimization is actually carried out.
> >
> > The comparison of the proposed method to conventional RQ and PQ methods is conducted in the R@1 setting, which is unusual. Typically, comparisons are made in more relaxed settings, such as R@5 or R@10. Furthermore, the proposed method is significantly slower than traditional approaches. In more relaxed settings, it is possible that the recall gap between the methods would narrow considerably (while the decoding time would still be much slower).

---

> > > ### Author Response · Authors · 2024-11-26
> > >
> > > **(1) “ Both this paper and the original QINCo paper leave the optimization process of the model somewhat unclear”**
> > > While we follow the optimization approach of QINCo [1], we agree with the reviewer that our paper would benefit from a more detailed overview of this process. We include this more detailed description of the optimization in Section A.2 “Training QINCo2” of the appendix, and provide these details below.
> > >
> > > The optimization problem defined by our loss (Eq. 1) can be expressed as $\arg\min_{\{C^M\},\{\theta_m\}} \mathbb E_x[\arg\min_{\{c^m \in C^m\}}        \mathcal{L}(x, F_\theta(c^1,\dots,c^M))]$, with $F_\theta$ representing a sequence of $M$ QINCo networks. The external optimization problem is fully differentiable, while the inner optimization problem corresponds to the encoding process (Sec 3.2).
> > > Following QINCo [1], we we alternate between solving the inner problem (encoding process) for a batch of data, and jointly optimizing all the $\theta^m$ and $C^M$ using a gradient-based optimizer. Our only changes to this procedure (Sec 3.2) are 1) we replace the RQ encoding process by our own (inner problem), and 2) we use a different gradient-based optimizer (AdamW).
> > >
> > > [1] Residual Quantization with Implicit Neural Codebooks. Huijben et al., ICLM 2024.
> > >
> > >
> > > **(2) Metrics of comparison**.
> > > In the revised manuscript we already added experimental results comparing to other methods in terms of R@10 and R@100, see the paragraph “Retrieval accuracy with relaxed settings” on page 16, and Table S4 on page 17 for the results. These results correspond to the setting of search among 1M vectors, as in Table 3 of the main paper.
> > > The differences between methods indeed narrow for these relaxed metrics, which is expected precisely because the metric is more relaxed. Ultimately, with sufficient bitrate and for large enough k, all R@k metrics converge to 100 for any method, which we can already observe happening for R@100 for some datasets in Table 17.

---

### Official Review · Reviewer_WeRp · 2024-11-03

**Soundness:** 4
**Presentation:** 4
**Contribution:** 3
**Rating:** 6
**Confidence:** 4

**Summary:**

QINCo2 is a deep-learning based vector quantizer that improves off of QINCo. The basic idea of both is to extend the idea of residual quantization (RQ) via deep learning. RQ is a greedy approach that quantizes a vector by doing each successive codeword selection to minimize the assignment loss so far. The QINCo family of quantizers adds a neural network that adapts the current codeword depending on the quantized representation so far, i.e. if $\hat{x}_i$ is the quantized representation of $x$ after $i$ codes, RQ does $\hat{x}_i=\hat{x}\_{i-1}+c_i$ while QINCo does $\hat{x}_i=\hat{x}\_{i-1}+f(c_i,\hat{x}\_{i-1})$ with learned $f$.

The main improvements from the original QINCo are:
1. Faster encoding by leveraging a faster, approximate $f$ to generate initial quantization candidates, and only re-ranking the top candidates with the full $f$.
1. Beam search during encoding, to make up for quality loss from approximate $f$ above.
1. Slight tweaks to model architecture and training hyperparameters.
1. Using a pairwise codebook procedure during decoding so that the vanilla additive decoder more closely resembles QINCo's implicit codebook results.

**Strengths:**

1. Figures are well-crafted and make the paper easy to understand
1. Extensive empirical results that break down the effect on quantization quality and encode/decode time for each adjustment relative to QINCo

**Weaknesses:**

1. Lack of source code release: considering these are fairly small models trained on open datasets, releasing code for reproducibility shouldn't have been difficult.
1. Limited novelty: this work only only suggests a minor change to the QINCo idea.

**Questions:**

1. A detailed description of an ANN use case that clearly benefits from QINCo2 would strengthen this paper. This paper currently shows that QINCo2 outperforms other quantizers at iso-bitrate in terms of quantization error, but pays more in terms of decoding cost. It could perhaps be argued that using other quantization methods to compress the vectors, and storing such compressed data on a cheaper storage medium (ex. flash) could perhaps beat QINCo2 in both storage cost and decoding cost. Quantifying whether or not this is the case would be very useful.
1. Source code?

---

> ### Author Response · Authors · 2024-11-21
>
> We are happy the reviewer finds our paper “easy to understand” and refers to our experiments as “Extensive”. Let us address the comments in the weaknesses and questions section below.
>
> **W1+Q2 (“Source code”)** The source code will be released upon acceptance of the paper, and will include the model weights used for our results.
>
> **W2 (“Limited novelty”)** We believe the work presented in our paper does have significant novelty, in particular the candidate pre-selection and pairwise approximate decoder are conceptually novel. Together with the use of beam search, this leads to runtime-controllable parameters to navigate the compute-accuracy tradeoff for neural decoders. These innovations together contribute to our substantially improved results w.r.t. the state-of-the-art results of QINCo, improving reconstruction MSE by 44% for 16-byte vector compression on BigANN, and search accuracy by 24% with 8-byte encodings on Deep1M.
>
> **Q2 (“use case that clearly benefits from QINCo2”)** Please refer to our general response to reviewers, where we address this comment together with similar ones from other reviewers.

---

> > ### Comment · Reviewer_WeRp · 2024-11-24
> >
> > 1. Source code: while I hope this to be the case, I will review based on the material I have at hand, so this does not sway my opinion.
> > 1. The ideas you listed may not have appeared in the original Qinco, but they have all been explored in existing quantization / source coding works. But novelty is somewhat a matter of opinion so perhaps this isn't worth discussing further.
> > 1. I don't see the general response as particularly strong, because it doesn't concretely show a use case where high compression is needed. The same QPS / dataset size parameters could be served with less bit-efficient (but more CPU-efficient) quantization schemes stored on a cheaper medium (ex. flash instead of RAM).

---

> > > ### Author Response · Authors · 2024-11-26
> > >
> > > **A3 (“I don't see the general response as particularly strong”)**: we study the frontier for quantization in a specific setting, specifically under a fixed memory budget using a single storage medium (RAM). While we agree with the reviewer that in certain application scenarios, the monetary costs could be balanced to change methods and constraints, we do not believe that it could be used as an argument against our method.
> > >
> > > We also respectfully disagree with the reviewer about the statement that choosing a cheaper medium (such as flash) would necessarily benefit more for RQ than for QINCo2. Such memories add an additional latency because of slower memory access. Compared to QINCo2 on RAM, we do not have any assurance that it would effectively be faster. Moreover, QINCo2 could also use such storage. First, an increased memory latency would reduce the impact of the CPU operation in the timing. Second, as shown in the newly added paragraph “Latency” at the end of App. B), QINCo2 retrieves less samples from memory for similar operating ranges, which would be at the advantage of our method on such a setting. Additionally, retrieval on large-scale databases with low QPS, fitting our operating SOTA range, will have lower CPU costs and higher memory costs than high QPS settings, which clearly benefits from our method, whatever the choice of storage is. Overall, we think that off-RAM retrieval is a different setting, and while interesting, it can’t be used as an argument against our method, and we have no indication that QINCo2 would perform less favourably in such a setting than RQ.

---

> > > > ### Comment · Reviewer_WeRp · 2024-11-26
> > > >
> > > > I was not making the point that cheaper storage mediums benefit other quantization schemes more. I was trying to make an apples-to-apples comparison. For fixed recall:
> > > > * Traditional quantization schemes would use more storage and less CPU per byte read
> > > > * The proposed method uses less storage and more CPU per byte read
> > > >
> > > > So by putting traditional quantization schemes on flash, it makes storage cheaper (by ballpark estimates, almost surely cheaper than the proposed method on RAM), and so if they also provide higher QPS with the same CPU, these traditional schemes would strictly dominate the proposed method.
> > > >
> > > > The latency overheads of using flash and having to read more data (largely irrelevant in flash, since read granularity is so coarse anyways) could easily be less than the CPU overhead of the proposed method. So ultimately, with no experiments to prove otherwise, the proposed method doesn't improve ANN SOTA.

---

> > > > > ### Author Response · Authors · 2024-11-26
> > > > >
> > > > > It would indeed be interesting to explore hybrid-memory approaches, since they move the tradeoff quite a bit. For example, what would happen if we allowed CPU RAM + GPU RAM, where the compute / storage tradeoff is even more in favor of compute ? We can mention hybrid approaches in future work.
> > > > >
> > > > > However, the point that we make in the paper is that in the RAM-only setting that we focus on, QINCo is the only method that reaches a wide range of high-precision operating points.

---

### Official Review · Reviewer_SSVZ · 2024-11-04

**Soundness:** 3
**Presentation:** 3
**Contribution:** 3
**Rating:** 6
**Confidence:** 3

**Summary:**

The paper presents QINCO2, an advanced method for vector compression and large-scale nearest neighbor search, building on the QINCO framework. QINCO2 introduces several key enhancements to improve the efficiency and accuracy of vector quantization, including: (i) QINCO2 incorporates codeword pre-selection and beam search, which improve encoding precision without exhaustive evaluations of all codebook options; (ii) an approximate decoder based on codeword pairs; (iii) an optimized training approach. The paper validates QINCO2's performance on datasets such as BigANN and Deep1M, demonstrating substantial improvements.

**Strengths:**

1. QINCO2’s use of beam search for vector encoding and codeword pre-selection represents a significant advancement over previous methods, optimizing both encoding time and quantization accuracy.
2. The introduction of a fast, approximate decoder based on codeword pairs offers a novel solution to the computational challenges of large-scale vector search, enhancing speed without a major sacrifice in accuracy.
3. The paper conducts thorough empirical evaluations across multiple datasets, showing substantial reductions in mean squared error (MSE) for vector compression and improvements in search accuracy compared to the original QINCO and other state-of-the-art models.

**Weaknesses:**

1. It would be beneficial to compare QINCO2 with other non-uniform quantization methods. Can QINCO or QINCO2 be extended to work with other large language models (LLMs), such as the LLaMA family?
2. The inference time remains high, especially in large-scale applications.
3. This method requires multiple heuristics and iterative steps to reach an optimal solution, which makes it appear more like a refinement rather than a groundbreaking improvement over QINCO. Including more mathematical analysis or theoretical proofs would strengthen the approach.
4. In line 205, you mention that "$g$ uses the same architecture as $f$." Did you experiment with alternative architectures for $g$?
5. In Figure 2, you note "Keep A candidates for each beam." Did you consider keeping a single candidate set for multiple beams?

**Questions:**

Please refer to Weakness.

---

> ### Author Response · Authors · 2024-11-21
>
> We would like to thank the reviewer for underlining the “novel solution” our work offer and the “represents a significant advancement” it represents, shown by our “empirical evaluations across multiple datasets”. We would like to address the reviewer concerns and questions below:
>
> **W1 (“other non-uniform quantization methods … extended to work with other large language models (LLMs), such as the LLaMA ”)** We weren’t quite sure to fully understand this question from the reviewer. So please let us know if our answers do not provide the information the reviewer was looking for. \
> (a) QINCo2 as well as methods we compare to (PQ, RQ, UNQ, QINCo) all rely on non-uniform quantization with k-means, but embedded in different quantization pipelines. \
> (b) Regarding extension to “other LLMs”, in our experiments, QINCo2 is used to quantize text embeddings from the Contriever model. Decoder-only large languages such as the LLaMA family do not clearly provide embeddings for text. If embeddings were nonetheless extracted from such a model, then yes, QINCo2 could be used to quantize them.
>
> **W2 (“high inference time”)** Please refer to our general response to reviewers, where we address this comment together with similar ones from other reviewers.
>
> **W3 (“multiple heuristics and iterative steps”)** We weren’t quite sure what the reviewer was pointing to precisely. Please let us know if our response does not address your concern.\
> (a) Regarding our large-scale search approach: Integration of a neural decoder into a billion-scale nearest neighbor search pipeline requires intermediate steps to balance speed and precision as described in Section 3.2. Such multi-stage pipelines are customary in other vector search works. See e.g. Huijben et al., ICML’24, section 3.3 in Morozov&Babenko, ICCV’19 and section 3.4 in “Automating Nearest Neighbor Search Configuration with Constrained Optimization” by Sun, Guo, Kumar, ArXiV’19. But the core of our contribution, the improved quantization algorithm, does not rely on this pipeline, as described in detail in Section 3.2 and experimentally validated in Section 4.2. \
> (b) If the reviewer is pointing to Appendix A2 where we list the training details for QINCo2: our goal here was to aim for maximal transparency about all settings used to train our models, and we note that these include mostly hyper-parameter choices (such as batch size, learning rate, optimizer, etc) that have to be made anyway.
>
> **W4 (“alternative architectures for $g$”)** In preliminary experiments we explored other architectures for codeword pre-selection, including linear projections, combinations with additive and multiplicative components, etc. We retrained the solution described in L205-207 where we just use a (second) trained codebook for preselection, i.e. with $g(c|x) = c$, as we found it to provide the best compute-accuracy tradeoff as compared to more complex alternatives. We compared it with using smaller models with the same architecture as they were particularly effective and the function $g(c|x) = c$ can be expressed as a special case of smaller models by setting depth to 0.
>
> **W5 (“keeping a single candidate set for multiple beams”)** Using the same pre-selected candidate codewords for each hypothesis in the beam will generally lead to worse results (as the candidates are no longer selected for optimality per hypothesis). Moreover, when using shared candidates, we would still need to evaluate the neural network f(c,x_hat) for each candidate for each hypothesis – i.e. $A\times B$ evaluations –, as QINCo adapts codeword c based on the partial reconstruction x_hat (specific to each hypothesis). Therefore sharing the candidate across hypotheses would not lead to speed improvements.

---

> > ### Comment · Reviewer_SSVZ · 2024-11-26
> >
> > Thanks Author for the detailed response. After thoroughly reviewing your responses and other responses, I stand by the positive score.

---

### Official Review · Reviewer_sWLT · 2024-11-04

**Soundness:** 2
**Presentation:** 2
**Contribution:** 2
**Rating:** 6
**Confidence:** 4

**Summary:**

This paper introduces a variant of QINCo which predicts codebooks per step according to the previous encode part. QINCov2 develops many tricks such as a better training procedure, beam search, etc., to improve its performance. Extensive experiments across multiple benchmark datasets demonstrate its superior performance.

**Strengths:**

- The proposed method achieves state-of-the-art performance on several benchmarks
- Extensive experiments demonstrate the effectiveness of each component.

**Weaknesses:**

- The task scenarios are not convincing. Previous work shows that QINCo [1] has significantly lower encoding and decoding speeds than PQ and RQ, and there is no obvious improvement in the paper. Figure 6 also shows nearly an order of magnitude less QPS than PQ/RQ in the low recall region. The authors should provide more explanation of why improving accuracy at the cost of QPS is necessary.
- Latency comparison with other methods is not considered in experiments.

**Questions:**

- Figure 6 demonstrates the retrieval accuracy/efficiency trade-off, but only R@1 is considered. How would the QPS/task accuracy trade-off be affected if a re-rank stage is added to RQ and PQ with relaxed settings such as R@10?
- Figure 4 only demonstrates the encoding/decoding speed of QINCov2. It is recommended to provide a more comprehensive comparison with QINCo, etc., similar to Table 3 in [1].
- It is advised to add a latency comparison of the full retrieval pipeline with other methods.

[1] Residual Quantization with Implicit Neural Codebooks

**Details Of Ethics Concerns:**

None.

---

> ### Author Response · Authors · 2024-11-21
>
> We would like to thank the reviewer for acknowledging that we report “state-of-the-art performance on several benchmarks” and that “Extensive experiments demonstrate the effectiveness of each component”.
> Below we address the different points raised in the review.
>
> **W1 (“task scenarios are not convincing”)** Please refer to our general response to reviewers, where we address this comment together with similar ones from other reviewers.
>
> **W2+Q3 (latency)** We agree with the reviewer that it is interesting to also consider the latency of different methods when processing a single query. We note that QINCo2 is implemented in python, and that the same holds true for QINCo, as compared to c++ implementations for PQ and RQ. To compare latencies we selected the operating point for BigANN1B 16 and 32 bytes where QINCo2 and IVF-RQ have approximately the same QPS & R@1.
> On BigANN1B (16 bytes) at a point with R@1=37 and QPS=2700, RQ has a latency of 10.78ms, and 9.10ms for QINCO2. On BigANN1B (32 bytes) at a point with R@1=62 and QPS=350, RQ has a latency of 71.54ms, and 22.25ms for QINCO2. While RQ uses larger (computationally more expensive) parameters for the Faiss query to achieve this accuracy, QINCo2 uses a less accurate and faster Faiss setting combined with a precise QINCo2 re-ranking. The smaller latency for QINCo2 might indicate that the Faiss search benefits more from batched queries, and that QINCo2 can bring substantial improvements to speed with a small number of queries. We thank the reviewer for this interesting suggestion, and we added these results to the paper.
>
> **Q1 (adding rerank stage to PQ/RQ for search, considering relaxed metrics such as R@10)**\
> (a) Re-ranking PQ/RQ search results using additional finer codes or original data would improve accuracy at the cost of additional storage, and therefore does not fit the (standard) experimental setting with constant memory budget. We note also that in the 2021 BigANN approximate nearest-neighbor search benchmark, searching in-RAM and in RAM+flash (which is a typical setting for reranking with a more precise representation) are entirely different tracks, which highlights that these settings are not comparable.\
> (b) When using less restrictive metrics such as R@10 instead of R@1, we observe that all methods get closer to each other in accuracy while maintaining the ordering of results. Therefore we found it more useful to report the R@1. For sake of completeness we added Table S4 to the supplementary material where we report R@10/100 for the experiments in Table 3.
>
> **Q2 (encoding/decoding speed of QINCov2)**
> The encoding times in Figure 4 do not compare to related work as they correspond to ablations specific to QINCo2. In L231-232 we express the encoding and decoding complexity of QINCo2 in terms of the main parameters (vocabulary size, number of encoding steps, etc.), but we agree with the reviewer that a comprehensive comparison to previous approaches, akin to Table 3 in the QINCo paper would be useful. We added it to the paper as Table S2 in the supplementary material. Compared to the small QINCo1(L=2) version reported in the table, our QINCo2-S model has a slower encoding (2910μs instead of 823μs) and a faster decoding (6.2μs instead of 8.3), which aligns with our goal of prioritizing decoding speed.

---

> > ### Author Response · Authors · 2024-11-28
> >
> > We have done our best to address the reviewer's concerns. We would like to ask the reviewer if our answers, additional results and updated paper addressed his concerns about our paper and the relevance of our setting. If any point remains unclear, we are available to discuss those concerns.

---

> > > ### Comment · Reviewer_sWLT · 2024-12-02
> > >
> > > Thank you for the rebuttal. The response basically addressed my concerns. After considering the feedback from the other reviewers, I have decided to increase my score.

---

### Comment · Area_Chair_8Ldq · 2024-11-13
**authors - reviewers discussion open until November 26 at 11:59pm AoE**

Dear authors & reviewers,

The reviews for the paper should be now visible to both authors and reviewers. The discussion is open until November 26 at 11:59pm AoE.

Your AC

---

### Author Response · Authors · 2024-11-21

We would like to thank the reviewers for their insightful remarks and encouraging comments on our work. Reviewers refer to our experimental evaluation as “extensive” or “thorough” (sWLT, SSVZ, WeRp, tRnB), and the comment on the obtained results as “SOTA” (sWLT), and improvements over prior work “significant” (SSVZ, xDMj, tRnB). Although one reviewer found the paper “hard to follow” (xDMj) two others comment that our paper is “well written” and “easy to understand” (tRnB, WeRp).

We have updated the manuscript following the suggestions in the reviews, and marked the additions in blue to facilitate finding these passages.

We respond to other specific comments point-by-point in a response to individual reviewers.
Before that, we would like to address a point raised by multiple reviewers.

**Common answer to**
- **sWLT: W1 “task scenarios are not convincing”**
- **SSVZ: W2 “inference time remains high”**
- **WeRp: Q1 “use case that clearly benefits from QINCo2”**

QINCo and QINCo2 function at operating points where the compression rate is very high. The drawback is that the vector search is slower than the baseline methods IVF-PQ and IVF-RQ, as shown in Figures 6 and S1. However, we argue that most of the commercial search engines have offerings for large datasets with a relatively low query rate (=Queries Per Second). For example, Pinecone is one of the major vector search engines, and their available offerings are between 10 and 150 QPS for 1M vectors (https://docs.pinecone.io/guides/indexes/choose-a-pod-type-and-size#queries-per-second-qps). In contrast, our operating points are around 500 QPS for 1B vectors. This is certainly on different hardware, and Pinecone’s online service probably has other overheads, but it illustrates that there are applications for low-QPS and high-compression operating points.

---

### Comment · Area_Chair_8Ldq · 2024-11-25

Dear reviewers,

The authors have provided individual responses to your reviews. Can you acknowledge you have read them, and comment on them as necessary? The discussion will come to a close very soon now:
- Nov 26: Last day for reviewers to ask questions to authors.
- Nov 27: Last day for authors to respond to reviewers.

Your AC

---

### Meta-Review · Area_Chair_8Ldq · 2024-12-19

**Metareview:**

This paper is slightly above borderline in terms of scores, and all reviewers suggested acceptance, although without much enthusiasm. My overall impression from the paper itself and the reviews and discussion is that this is a good piece of engineering work (apart from a few issues noted in the reviews), but somewhat derivative, in that it is an incremental improvement (version 2, as the "Qinco2" title says) over a "Qinco" approach published just a few months ago. The improvements appear to be the addition to Qinco of standard techniques or minor refinements (such as "an optimized training procedure and network architecture"). The gain in performance is decent, at least in certain regimes of compression/search problems.

**Additional Comments On Reviewer Discussion:**

N/A

---

### Decision · Program_Chairs · 2025-01-22

Accept (Poster)